# CRISPR activation screen in mice identifies novel membrane proteins enhancing pulmonary metastatic colonisation

Louise van der Weyden [1], Victoria Harle[1,7], Gemma Turner[1,7], Victoria Offord[1], Vivek Iyer[1], Alastair Droop [1], Agnieszka Swiatkowska[1], Roy Rabbie [1], Andrew D. Campbell [2], Owen J. Sansom [2,3], Mercedes Pardo [4], Jyoti S. Choudhary [4], Ingrid Ferreira [1], Mark Tullett[5], Mark J. Arends [6], Anneliese O. Speak [1] & David J. Adams [1✉]

Melanoma represents ~5% of all cutaneous malignancies, yet accounts for the majority of skin cancer deaths due to its propensity to metastasise. To develop new therapies, novel target molecules must to be identified and the accessibility of cell surface proteins makes them attractive targets. Using CRISPR activation technology, we screened a library of guide RNAs targeting membrane protein-encoding genes to identify cell surface molecules whose upregulation enhances the metastatic pulmonary colonisation capabilities of tumour cells in vivo. We show that upregulated expression of the cell surface protein LRRN4CL led to increased pulmonary metastases in mice. Critically, *LRRN4CL* expression was elevated in melanoma patient samples, with high expression levels correlating with decreased survival. Collectively, our findings uncover an unappreciated role for LRRN4CL in the outcome of melanoma patients and identifies a potential therapeutic target and biomarker.

[1] Wellcome Sanger Institute, Wellcome Genome Campus, Hinxton, Cambridge, UK. [2] Cancer Research UK Beatson Institute, Glasgow, UK. [3] Institute of Cancer Sciences, University of Glasgow, Glasgow, UK. [4] Institute of Cancer Research, London, UK. [5] Western Sussex NHS Foundation Trust, Chichester, West Sussex, UK. [6] University of Edinburgh Division of Pathology, Edinburgh Cancer Research UK Cancer Centre, Institute of Genetics & Molecular Medicine, Edinburgh, UK. [7] These authors contributed equally: Victoria Harle, Gemma Turner. ✉email: da1@sanger.ac.uk

Metastasis is the spread of cancer cells to a secondary site and is the leading cause of death in cancer patients. Metastatic dissemination of a tumour is a multi-step process and involves tumour cell invasion of the lymphatics and/or blood vessels, survival in the circulation and extravasation at a distant site. Following this, the disseminated tumour cells (DTCs) must proliferate and colonise the secondary site to become an overt, clinically relevant metastasis. In vivo video-microscopy has shown that the early steps of the metastatic process are relatively efficient, and it is the regulation of DTC growth at the secondary site that determines metastatic outcome[1]. Thus, colonisation is a critical and rate-limiting step in the formation of clinically relevant metastases. The lungs are common sites of metastasis across many different cancer types, in part due to the fact that blood and lymphatic fluids returning from the periphery are pumped by the heart through the pulmonary microvasculature. More than 80% of metastatic melanoma patients initially show the involvement of only one distant organ site, which is most commonly the lungs[2]. Thus, controlling metastatic colonisation in the lungs is of critical importance for patient management.

The importance of cell surface proteins in the metastatic process is well-established, as they play critical roles in both signalling and adhesion interactions between tumour cells and the microenvironment. For example, there are cell-surface proteins that are functionally linked to tumour cell intravasation[3] and others that specifically induce homing to the lung microvasculature[4]. Cell surface proteins are also attractive drug and immunological targets due to their accessibility such as the transmembrane oncoprotein HER2 (encoded by *ERBB2*), which is overexpressed in ~20% of invasive breast cancers and can be specifically targeted with the humanised IgG1 antibody, Trastuzumab (Herceptin®)[5]. More recently, a study used quantitative surface proteomics to reveal proteins upregulated on human cancer cells transformed with *KRAS^{G12V}*, and found that antibodies targeting the CUB domain-containing protein 1 (CDCP1), a single-pass transmembrane protein highly overexpressed in a diverse range of human cancers, could be used to deliver cytotoxic and immunotherapeutic payloads to RAS-transformed cancer cells in vivo[6,7]. Indeed, whilst cell surface molecules represent only ~22% of all proteins encoded in the human genome, many approved drugs target overexpressed/mutated cell surface molecules (or their natural ligands) to suppress aberrant signalling pathways[8,9].

Traditionally, mass-spectrometry-based proteomics has been used to identify cell surface proteins important for cancer and metastasis[3,6]. In this study, we applied CRISPR activation (CRISPRa) technology to screen a membrane protein guide RNA (gRNA) library using an experimental metastasis assay with melanoma cells in mice. This screen identified cell surface proteins whose upregulated expression in melanoma cells resulted in enhanced pulmonary metastatic colonisation, and thus represent potential therapeutic targets and biomarkers.

## Results

**Screening and identification of *LRRN4CL*.** To identify cell-surface proteins whose upregulation results in the enhanced ability of melanoma cells to colonise the lung, we performed an experimental metastasis assay in mice using the weakly metastatic B16-F0 mouse melanoma cell line. B16-F0 cells expressing dCas9 (B16-F0-dCas9) were virally transduced with a CRISPRa gRNA library targeting genes that encode membrane proteins ('m6' library; Fig. 1a). The library contained 11,225 gRNAs in total, and was composed of gRNAs targeting 2195 membrane protein-encoding genes (with five guides/gene) and 250 non-targeting 'control' gRNAs, cloned into the pCRISPRia_v2 vector backbone[10]. Wild-type mice ($n = 70$) were tail vein dosed with $5.5 \times 10^5$ cells

(representing a 50× coverage of the library), then randomised into two cohorts: 35 mice were collected after 4 h (to assess the ability of gRNA-carrying tumour cells to enter the lung) and 35 mice were collected at 19 days (when the mice started showing clinical signs of pulmonary tumour burden). Genomic DNA (gDNA) was extracted from the lungs of the mice (after saline perfusion) and high-throughput sequencing of the gRNAs performed to identify their representation at 4 h and 19 days, relative to that present in the transduced B16-F0-dCas9 cell population before injection (Supplementary Fig. 1a). While the gRNA representation present in the pre-injection cells was comparable with that in the plasmid library (considering all gene-targeting and non-targeting control gRNAs), the gRNA representation in the 4-h and 19-day lung samples was significantly decreased, with anyone mouse carrying only ~20% of the gRNAs present in the library (Supplementary Fig. 1b), suggesting that extravasation and subsequent survival in the lung was a major biological bottleneck. In addition, the 19-day lung samples had enrichment of a subset of gRNAs, suggesting positive selection for cells carrying these gRNAs during the growth of the metastases (Supplementary Fig. 1c). To identify gRNAs significantly enriched in these mice, we considered two approaches: the 'percentile ranking' (PR) approach, which was based on single gRNA abundance amongst multiple mice (gRNAs found in the top 98th percentile of abundance/mouse), and a 'JACKS' analysis[11], which considered the relative abundance of all five gRNAs for each gene amongst tumours collected from multiple mice. Using the PR approach, the top two genes (gRNAs) were *Lrrn4cl* and *Slc4a3* (Supplementary Table 1); using JACKS analysis, the top two genes were *Lrrn4cl* and *Tm4sf19* (Supplementary Table 2). An experimental metastasis assay was performed using B16-F0-dCas9 melanoma cells transduced with single gRNAs targeting each of these three genes separately. Targeting of each gene resulted in a significant enhancement of the ability of the cells to colonise the lung, relative to cells transduced with a pool of three non-targeting control gRNAs (Fig. 1b).

Given that *Lrrn4cl* (*Leucine-Rich Repeat Neuronal 4 C-Terminal Like*) was identified as a 'hit' in both analyses, and showed the strongest enhancement of pulmonary metastatic capabilities, we chose to further characterise this gene. Consistent with the phenotype observed when B16-F0 cells were virally transduced with the dCas9/CRISPRa system to upregulate *Lrrn4cl* expression, transfection of a *Lrrn4cl* cDNA-containing plasmid into B16-F0 cells also resulted in enhanced pulmonary metastatic colonisation (Fig. 1c, d). Mouse and human *Lrrn4cl/LRRN4CL* are two-exon genes (with the entire coding region contained in exon 2) that encode 239 and 238 amino acid proteins, respectively, which show 66% identity (Fig. 1e). *LRRN4CL* is predicted by UniProtKB to be a single-pass type I integral membrane protein with a signal peptide, large ECD, small transmembrane/helical domain and a short cytoplasmic tail (Fig. 1e). The extracellular portion of the protein is predicted to undergo N-linked glycosylation (at position N132 in human and at N132 and N174 in mouse) and contains a fibronectin type-III (FN3) domain, found in many proteins involved in ligand binding (Fig. 1e). PNGase F-digestion of B16-F0 cell lysates expressing a FLAG/streptavidin-tagged *Lrrn4cl* cDNA (LRRN4CL-FSA) caused a shift of the LRRN4CL-FSA band to a lower molecular weight, confirming that LRRN4CL is glycosylated (Supplementary Fig. 2a). Images of anti-LRRN4CL immunohistochemical staining of human tissues showed that in the tonsil there was a strong expression on the cell membranes (in a ring pattern) and in the colon, there was mostly cell membrane staining (with enhanced staining at the luminal surface of colonocytes) and some cytoplasmic staining (Supplementary Fig. 2b, c). Tissue expression of *LRRN4CL* from the NCBI BioProject database (human data from PRJEB4337[12]) showed only low levels of expression in normal tissues (the highest

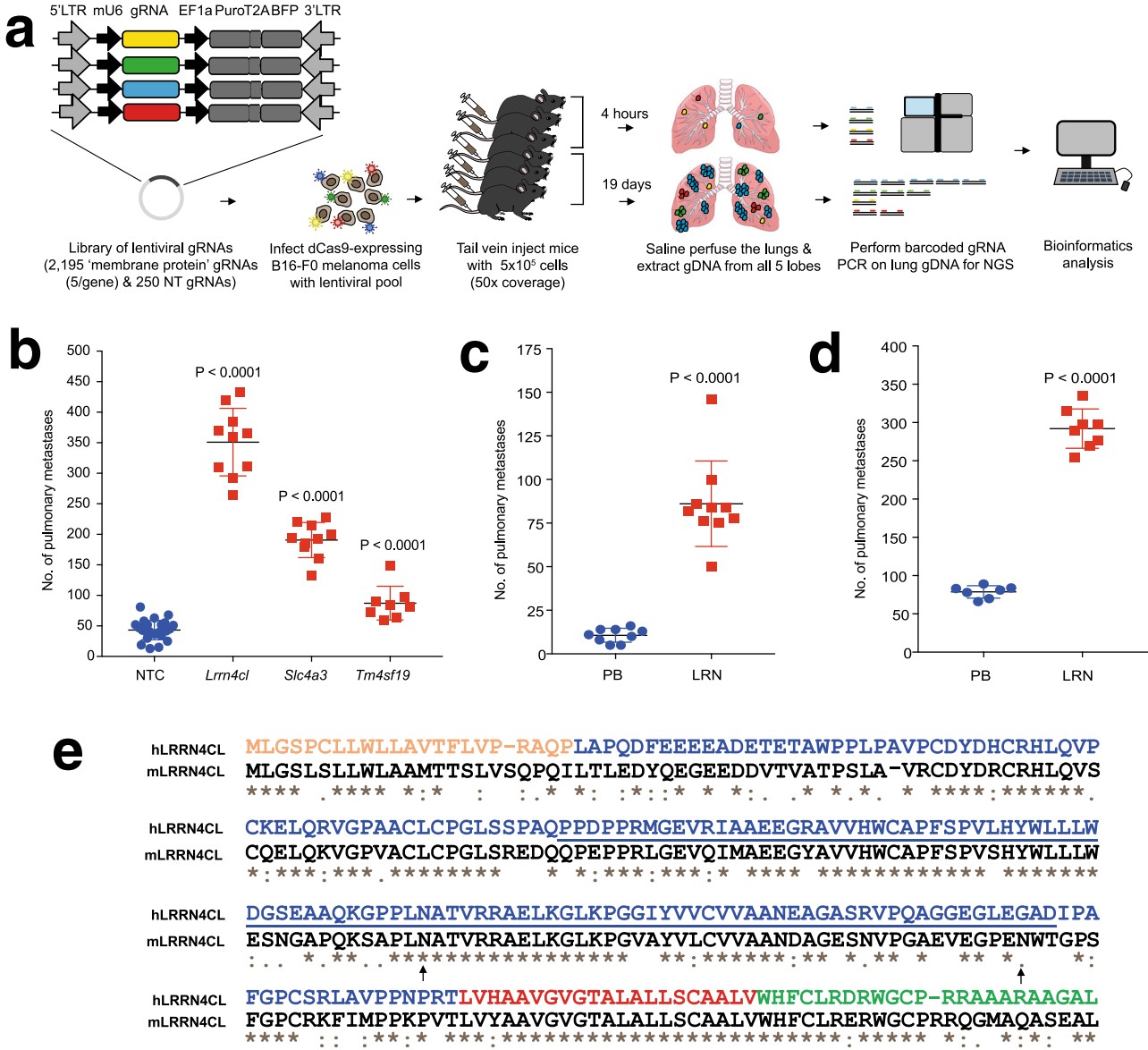

**Fig. 1 Performing a CRISPRa screen in vivo to identify cell surface regulators of pulmonary metastatic colonisation. a** Graphic representation of the screen outline. **b** The number of metastatic colonies in the lungs of mice 10 days after being tail vein dosed with $2 \times 10^5$ B16-F0-dCas9 cells carrying gRNAs against the genes listed. NTC, non-targeting control. **c** The number of metastatic colonies in the lungs of mice 10 days after being tail vein dosed with $1 \times 10^5$ B16-F0 cells that have been stably transfected with a plasmid carrying the *Lrrn4cl* cDNA (LRN) or an empty vector (PB). **d** The number of metastatic colonies in the lungs of mice 10 days after being tail vein dosed with $4 \times 10^5$ B16-F0 cells that have been stably transfected with a vector carrying the *Lrrn4cl* cDNA (LRN) or an empty vector (PB). For (b-d), each symbol represents a mouse, the bars represent mean ± SD, 2 independent experiments performed (representative data from one experiment is shown) and statistics performed using a Mann–Whitney *t* test. **e** The human and mouse LRRN4CL protein sequences (ENSP00000325808 and ENSMUSP00000093976, respectively) were aligned CLUSTAL W (1.81) in Ensembl. Below each site amino acid of the alignment is a key denoting conserved sites (\*), sites with conservative replacements (:), sites with semi-conservative replacements (.), and sites with non-conservative replacements (). The UniProtKB predicted location of the signal peptide (orange), extracellular domain (blue), transmembrane domain (red), cytoplasmic domain (green), FN3 domain (underlined) and N-linked glycosylation site at N132 (arrow) for human LRRN4CL are shown (glycosylation sites for the mouse also shown: arrows at N132 and N174).

expression is in the endometrium and ovary, with a mean of 5.9 RPKM for each) (Supplementary Fig. 2d), in agreement with proteomic data from the Human Proteome Map. We next generated and phenotyped *Lrrn4cl* knockout mice, finding them to be homozygous viable and fertile, in agreement with other studies[13] and displaying no gross differences compared with wildtype control mice in body weight, soft tissue mass, lean mass, fat mass, bone mineral content or bone mineral density (Supplementary Fig. 2e). Both of these factors are critical for LRRN4CL to be considered as a therapeutic target.

**Upregulated LRRN4CL expression confers enhanced lung-specific metastatic colonisation.** To confirm that the observed phenotype was not specific to B16-F0 melanoma cells, we expressed *Lrrn4cl* cDNA in three other mouse melanoma cell lines (HCmel12, YUMM1.7 and B16-BL6 cells [the latter of which already displays a high metastatic propensity]) and observed increased pulmonary metastatic colonisation relative to cells transfected with the empty vector alone (Fig. 2a). To assess whether the phenotype associated with *Lrrn4cl* upregulation was specific to melanoma cells, we expressed the *Lrrn4cl* cDNA in

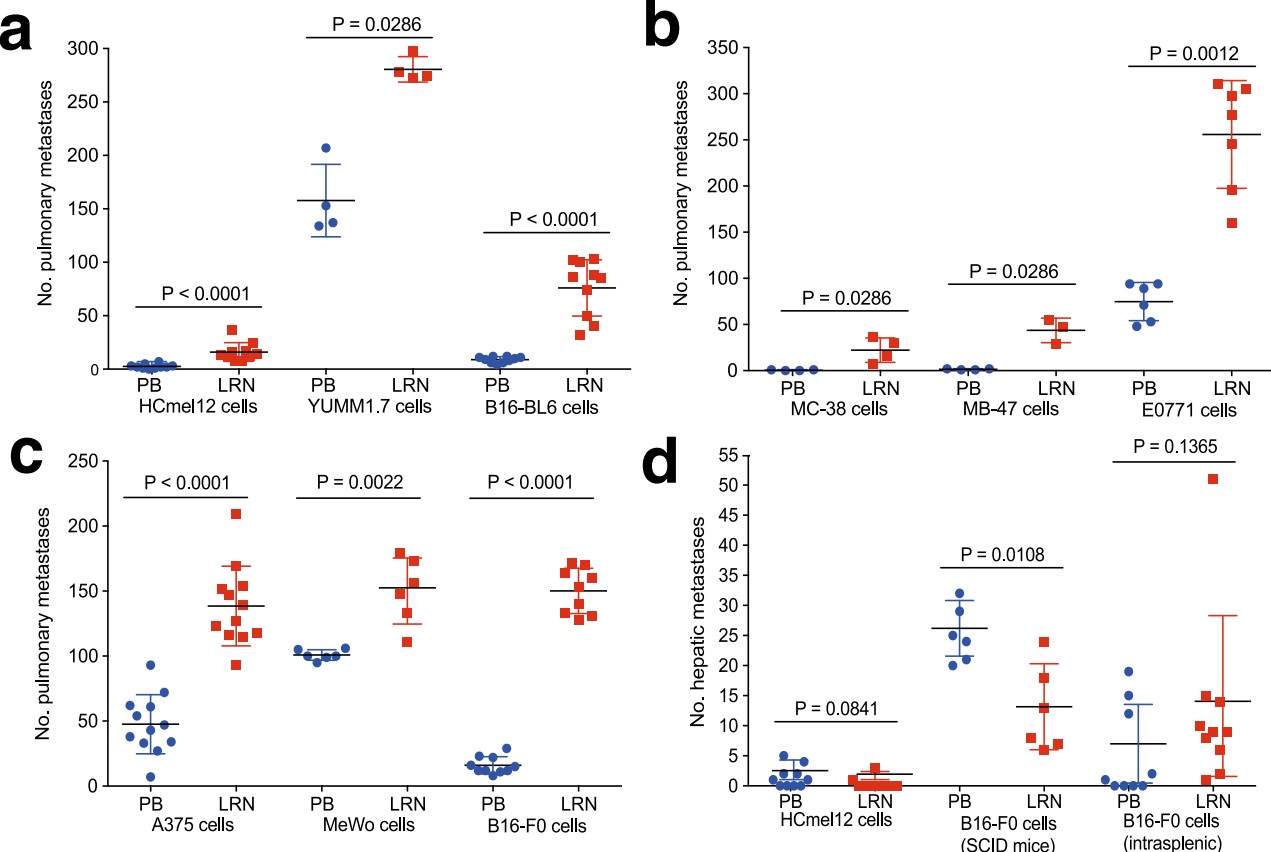

**Fig. 2 The effect of LRRN4CL over-expression on metastatic colonisation. a** The number of metastatic colonies in the lungs of mice 10 days after being tail vein dosed with melanoma cell lines ($5 \times 10^5$ HCmel12 cells, $4 \times 10^5$ YUMM1.7 cells and $0.25 \times 10^5$ B16-BL6 cells) that have been stably transfected with a vector carrying mouse *Lrrn4cl* cDNA (LRN) or an empty vector (PB). **b** The number of metastatic colonies in the lungs of mice 10 days after being tail vein dosed with a colorectal cell line ($4 \times 10^5$ MC-38 cells), bladder cancer cell line ($4 \times 10^5$ MB-49 cells), or breast cancer cell line ($4 \times 10^5$ EO771 cells) that have been stably transfected with a vector carrying mouse *Lrrn4cl* cDNA (LRN) or an empty vector (PB). **c** The number of metastatic colonies in the lungs of mice 30 days after being tail vein dosed with human melanoma cell lines ($2 \times 10^5$ A375 cells or $0.5 \times 10^5$ MeWo cells) or 10 days after being dosed with a mouse melanoma cell line ($4 \times 10^5$ B16-F0 cells) that have been stably transfected a vector carrying human *LRRN4CL* cDNA (LRN) or an empty vector (PB). For a–c, each symbol represents a mouse, the bars represent mean ± SD, two independent experiments were performed (representative data from one experiment is shown) and statistics were performed using a Mann–Whitney *t* test. **d** The number of metastatic colonies in the livers of mice after being tail vein dosed (with $5 \times 10^5$ HCmel12 cells (wild-type mice) or $1 \times 10^5$ B16-F0 cells (NOD-SCID mice)) or intrasplenic dosed ($0.1 \times 10^5$ B16-F0 cells (wildtype mice)) with melanoma cells that have been stably transfected a vector carrying mouse *Lrrn4cl* cDNA (LRN) or an empty vector (PB). Each symbol represents a mouse, the bars represent mean ± SD, and statistics were performed using a Mann–Whitney *t* test.

three non-melanoma mouse cancer cell lines (MC-38 colorectal cancer cells, MB-47 bladder cancer cells and EO771 breast cancer cells) and the same effect was observed (Fig. 2b). Furthermore, we performed an additional screen using tail vein dosed E0771-dCas9 cells transduced with the membrane protein (m6) CRISPRa gRNA library and collected the lungs of the mice when they showed signs of pulmonary tumour burden. Using the PR approach to identify gRNAs significantly enriched in these mice, we found the top hit in this screen was a gRNA targeting *Lrrn4cl* (the same one found in the B16-F0 screen; *Lrrn4cl_+_8850757.23-P1P2*). Indeed, all five gRNAs targeting *Lrrn4cl* were found in the 98th percentile of at least one mouse in this screen. Thus, although we identified *Lrrn4cl* in melanoma cells, it mediates pulmonary colonisation cell lines from other tumour types.

We next expressed the human *LRRN4CL* cDNA in two human melanoma cell lines, A375 and MeWo (Western blot confirming upregulated expression shown in Supplementary Fig. 3) and again observed increased pulmonary metastatic colonisation relative to cells transfected with the empty vector alone (Fig. 2c). Similarly, expression of human *LRRN4CL* cDNA in mouse B16-F0 cells

resulted in increased pulmonary metastatic colonisation (Fig. 2c), suggesting a conserved role in the regulation of metastatic colonisation.

Tail vein dosing of mouse HCmel12 melanoma cells primarily resulted in pulmonary metastases, however, extrapulmonary metastases were also observed in some mice, specifically within the liver (that was not observed with other cell lines). It was interesting to note that the expression of *Lrrn4cl* cDNA did not result in increased hepatic metastatic colonisation (Fig. 2d). Similarly, when mouse B16-F0 melanoma cells were tail vein dosed into immunodeficient (NOD-SCID) mice, in addition to pulmonary metastases, hepatic metastases were also noted, and expression of *Lrrn4cl* cDNA resulted in the cells showing significantly decreased hepatic metastatic colonisation (Fig. 2d). Intrasplenic injection of B16-F0 cells, which results in hepatic metastatic colonisation, showed no difference in the number of hepatic metastatic colonies between *Lrrn4cl* over-expressing cells and control cells (Fig. 2d). These data suggest that it is specifically pulmonary metastatic colonisation that is enhanced by *Lrrn4cl* and not metastatic colonisation in general.

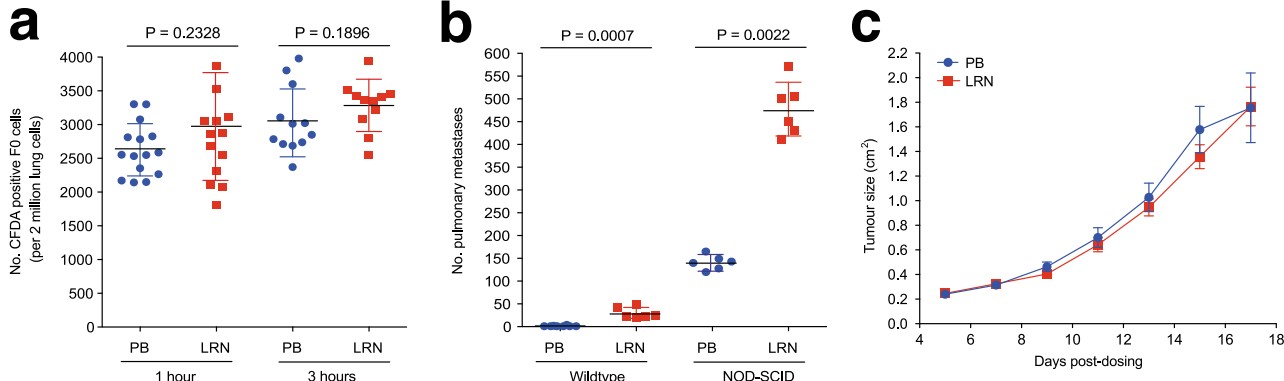

**Fig. 3 Investigation of the mechanism of action of LRRN4CL. a** The number of CFDA-positive B16-F0 cells present in the lung (per 2 million lung cells counted) after either 1- or 3-h post-tail vein dosing. The cells were stably transfected with a vector carrying mouse *Lrrn4cl* cDNA (LRN) or an empty vector (PB), and then labelled with the fluorescent tracker dye, CFDA. Each symbol represents a mouse, the bars represent mean ± SD, and statistics were performed using a Mann–Whitney *t* test. **b** The number of metastatic colonies in the lungs of either wildtype or immunodeficient (NOD-SCID) mice 10 days after being tail vein dosed with B16-F0 cells that have been stably transfected with a vector carrying mouse *Lrrn4cl* cDNA (LRN) or an empty vector (PB). Each symbol represents a mouse, the bars represent mean ± SD, two independent experiments were performed (representative data from one experiment is shown) and statistics were performed using a Mann–Whitney *t* test. **c** The growth of B16-F0 cells that have been stably transfected with a vector carrying mouse *Lrrn4cl* cDNA (LRN) or an empty vector (PB) after subcutaneous administration into the flank of wildtype mice. The data are shown as mean ± SEM with *n* = 8–9 mice per cell line per experiment. Two independent experiments were performed (representative data from one experiment is shown).

**Insight into the mechanism of action of LRRN4CL-mediated pulmonary colonisation**. To understand the mechanism of action of *Lrrn4cl*, we tail vein injected wildtype mice with fluorescently labelled B16-F0 cells (transfected with *Lrrn4cl* cDNA [F0_LRN] or empty vector backbone [F0_PB]). After 1 or 3 h, the lungs were flushed with saline to remove any blood and tumour cells from the blood vessels, and we assessed the number of tumour cells that had undergone extravasation into the lungs. As shown in Fig. 3a, there was no difference in the number of F0_LRN or F0_PB tumour cells present in the saline-rinsed lungs at either 1 or 3 h post dosing. This suggests that expression of the *Lrrn4cl* cDNA did not enhance the ability of the tumour cells to extravasate. To ascertain whether LRRN4CL expression aided evasion of NK, T and B cells, B16-F0 cells were tail vein dosed into immunocompromised (NOD-SCID) mice and, similar to results in the wildtype mice, F0_LRN cells had significantly increased pulmonary metastatic colonisation relative to F0_PB cells (Fig. 3b). This suggests that the *Lrrn4cl*-mediated phenotype was not primarily regulated by NK, T and B cells. Furthermore, *Lrrn4cl* expression did not affect the primary tumour growth of B16-F0 melanoma cells subcutaneously administered to wildtype mice (Fig. 3c).

*LRRN4CL* is a novel gene whose function has not yet been characterised, so we first sought to identify protein binding partners. Using cell microarray screening technology, we assessed the ability of the extracellular domain (ECD) of human LRRN4CL (amino acids 23-194, C-terminally Fc tagged) to bind to members of a cDNA library encoding >5500 full-length human plasma membrane and tethered secreted proteins, expressed on the cell surface of human HEK293 cells. We identified a positive interaction between the human LRRN4CL ECD (LRRN4CL_ECD) and CRTAC1 (human CRTAC1α, human CRTAC1β and mouse CRTAC1), and validated this using a flow cytometric assay (Supplementary Fig. 4a, b). Positive interactions were also observed between LRRN4CL_ECD and various members of the Fc gamma receptor family, however, these were expected interactions mediated by the Fc domains of the test Fc fusion protein and/or the detection antibody. Although *CRTAC1* is strongly expressed in the human lung, *Crtac1* is barely detectable in the mouse lung (Supplementary Fig. 4c–e). These data are in agreement with a recently published

study that shows the evolutionary divergence of lung cell types and expression patterns between mouse and human, with *CRTAC1* being an example of a gene that is expressed by human AT2 cells but not by mouse AT2 cells[14]. Tail vein dosing of F0_LRN cells in *Crtac1* null mice resulted in no difference in the number of pulmonary metastatic colonies compared with wildtype mice (Supplementary Fig. 4f). This suggests that, at least in mice, LRRN4CL does not mediate the enhanced metastatic pulmonary colonisation phenotype through binding to CRTAC1.

We next applied in vitro assays to examine whether LRRN4CL affected the following intrinsic properties of melanoma cells that are generally associated with increased metastatic potential: general growth rate in vitro, invasion and migration, epithelial–mesenchymal transition, matrix metalloprotease (MMP) production, and self-renewal. Upregulated expression of LRRN4CL in A375 cells had no effect on any of these characteristics (Supplementary Fig. 5), which suggested that either LRRN4CL promotes lung metastasis by mechanisms distinct from the aforementioned processes, or that LRRN4CL only mediated its effects when in the lung microenvironment.

Based on these data, we considered a potential role for LRRN4CL in the later stages of the metastatic process, specifically DTC survival and subsequent outgrowth in the lung parenchyma. As an upregulated expression of LRRN4CL only produced a phenotype in the in vivo context, we performed RNA-seq on A375 melanoma cells that were tail vein dosed and grown in the lungs of mice for 21 days, in addition to cells grown in vitro, to identify any differentially expressed genes between these two contexts. Principal component analysis (PCA) grouped the samples into four clusters (Supplementary Fig. 6a): control cells grown in vitro (VITRO_E), LRRN4CL over-expressing cells grown in vitro (VITRO_L), control cells from the mouse lung (VIVO_E) and LRRN4CL over-expressing cells from the mouse lung (VIVO_L). Two samples, 'VIVO_L1c' and 'VIVO_L3a', were outliers from the VIVO_L group and so were excluded from the analysis. Interestingly the expression of *LRRN4CL* in the control cells was increased in vivo, relative to in vitro (VITRO_E cells versus VIVO_E cells; Supplementary Fig. 6b). Using a log₂ fold change cut-off of (1, −1) and an adjusted *P* value of *P* < 0.01, there were 115 significantly differentially expressed genes (DEGs;

including nine non-protein-coding genes) between *LRRN4CL* over-expressing (A375_LRN) cells and control (A375_E) cells that were only found in the in vivo context (Supplementary Data 1). There were 75 down-regulated genes and 40 upregulated genes, grouped into eight clusters (Supplementary Fig. 7). Gene set enrichment analysis (GSEA), using two different programmes found the most statistically significantly downregulated pathways to be 'interferon signalling' ($P_{adj} = 0.0002$; Reactome) and 'interferon alpha' and 'interferon gamma' ($P_{adj} = 5.5 \times 10^{-5}$; Hallmark) (Supplementary Fig. 8a, b). The genes in these pathways were in clusters 6 and 7, which showed the most striking differences between the control cells and *LRRN4CL* over-expressing cells in vivo (Fig. 4). Using the 13 genes from the Reactome 'interferon signalling' pathway as a 'signature' in survival analysis, decreased expression of these genes correlated with worse survival of melanoma patients (TCGA cohort; Supplementary Fig. 8c). Of the DEGs that were significantly up-regulated, many have known roles in promoting metastasis, including *LOXL1*, *MMP28*, *HAS1* and *FPR1*. Thus, in the lung, the enhanced expression of LRRN4CL allows the cells to enter a pro-metastatic phenotype.

**LRRN4CL is highly expressed in melanoma and correlates with worse patient survival.** The TCGA database shows there is a range of *LRRN4CL* mRNA expression levels across 32 cancer types; expression is enriched in melanomas, with cutaneous melanoma and uveal melanoma showing the highest levels (Fig. 5a). Using the TCGA dataset, higher expression levels of *LRRN4CL* (top 25% quartile) significantly correlated with a poorer outcome (disease-specific survival) in cancer patients in general (all cancer types included in the analysis; Cox Log-rank $P = 2.2 \times 10^{-16}$, age- and sex-adjusted; Fig. 5b). In melanoma patients, high expression of *LRRN4CL* significantly correlated with worse outcome, in both the TCGA dataset (Cox Log-rank $P = 0.0008$, age- and sex-adjusted; Fig. 5c), and the AVAST-M dataset[15] (Cox Log-rank $P = 0.048$ [univariate analyses], Fig. 5d and Supplementary Fig. 9a, respectively). Interestingly, in uveal melanoma patients, higher expression of *LRRN4CL* did not correlate with clinical outcome (Supplementary Fig. 9b). Whilst this could possibly be due to the limited number of patients in this cohort ($n = 20$), it is important to note that uveal melanoma shows a strong preference for metastasis to the liver[16] (where we showed upregulated expression of *LRRN4CL* did not result in enhanced metastatic colonisation abilities; Fig. 2d). Analysis of the TCGA datasets also showed that high expression of *LRRN4CL* correlated with worse outcome in low-grade glioma, renal clear cell carcinoma and ovarian carcinoma (univariate analysis, Log-rank $P = 1.17 \times 10^{-6}$, 0.027 and 0.048, respectively; Supplementary Fig. 9c–e).

## Discussion
To identify cell-surface proteins whose upregulation results in an enhanced ability of melanoma cells to colonise the lung, we performed an in vivo CRISPRa screen in mice using B16-F0 mouse melanoma cells transduced with a library of 'membrane protein' CRISPRa gRNAs (Fig. 1a). We used an experimental metastasis assay that focuses on later stages of the metastatic cascade, specifically tumour cell extravasation, and survival and subsequent proliferation at the secondary site to become overt metastases[17]. We chose to use B16-F0 cells for the initial screen as we and others have previously shown that they are very weakly metastatic[18] and thus would lead to less 'background' in the screen, i.e., cells that will enter the lung and proliferate, regardless of the gRNA they are carrying, due to their inherent ability to metastasise. Analysis of the lungs at 4 h post dosing with tumour

cells revealed that only ~20% of the total number of gRNAs were present in any one mouse, although the whole library was represented across the cohort of 35 mice (Supplementary Fig. 1). Such a strong biological bottleneck is consistent with previous reports of tail vein administration of the related weakly metastatic B16-F1 melanoma cells, in which only ~64% of the cells were present in the lung 2 min post dosing, and only ~32% after 3 h[19].

The CRISPRa system has been successfully used in vivo for screening pools of gRNAs targeting transcription start sites of 1–2 genes[20,21], however, as yet there have been no large-scale in vivo screens performed using this system. Thus, we utilised two different approaches by which to identify genes (gRNAs) enriched in the lungs of mice at 19 days post dosing. The first method, similar to that used in a recent genome-scale in vivo Cas9/CRISPR screen in T cells[22], was to rank the individual gRNAs based on their relative abundance within each mouse (at the 19-day time point) and identify those in the top 98th percentile. In this analysis, the majority of enriched gRNAs were present in ≤2 mice, however, two gRNAs were identified in 4 mice and 3 mice, which targeted *Lrrn4cl* and *Slc4a3*, respectively. *Lrrn4cl* encodes a protein of unknown function and *Slc4a3* (*solute carrier family 4 members 3*) encodes a plasma membrane anion exchange chloride/bicarbonate transporter (also known as AE3) that has been shown to be involved in cellular transformation, with upregulated expression affecting cell viability and cellular attachment[23]. The second method was to use 'JACKS' (Joint Analysis of CRISPR/Cas9 Knockout Screens)[11] which is a Bayesian method that models variable gRNA efficacies and thus considers all five gRNAs per gene during hit identification. Using this analysis, ranking the output by the mean of the JACKS score at 19 days minus the mean of the JACKS score at 4 h, the top two genes were *Lrrn4cl* and *Tm4sf19*. *Tm4sf19* (*transmembrane 4L six family member 19*, also known as *Octm4*) is a member of the four-transmembrane L6 superfamily, which includes *TM4SF1*, whose overexpression in many cancers is associated with poor prognosis, and has roles in proliferation, invasion, and metastasis[24]. However, little is currently known about *TM4SF19*, although it was recently shown that lipopolysaccharides could upregulate TM4SF19 expression, resulting in reduced VE-cadherin expression and weakened endothelial cell adherens junctions[25]. Given that *LRRN4CL* was the top 'hit' of both analyses, we focused our efforts on this gene.

LRRN4CL is predicted to be a single-pass type I transmembrane protein. We have shown here that upregulated expression of mouse and human LRRN4CL resulted in an enhanced ability of metastatic tumour cells (both melanoma and other cell types) to colonise the lungs in mice (Figs. 1 and 2a–c). This phenotype was not due to an increased ability to extravasate from the circulation to the lung (Fig. 3a), nor an ability to avoid NK cells, T cells and B cells (Fig. 3b). The enhanced metastatic colonisation abilities of LRRN4CL appeared lung-specific, as the only extra-pulmonary metastases observed were in the liver and LRRN4CL over-expression did not increase hepatic colonisation (Fig. 2d). We did not observe any phenotypic differences compared with wildtype cells in terms of in vitro metastatic capabilities (Supplementary Fig. 5). In addition, upregulated expression of LRRN4CL in melanoma cells did not result in any phenotypic difference, compared with control cells, in primary tumour growth in the skin after subcutaneous administration into the flank (Fig. 3c).

As upregulated *LRRN4CL* expression only provided an advantage to metastatic tumour cells in the lung, with no phenotypic effect observed in vitro, we performed RNAseq of human A375 melanoma cells expressing LRRN4CL (A375_LRNmCherry cells; 'L') that had been colonising the lung for 21 days. We looked for differentially expressed genes, relative to control cells (A375_EVmCherry cells; 'E') after removing any genes that were

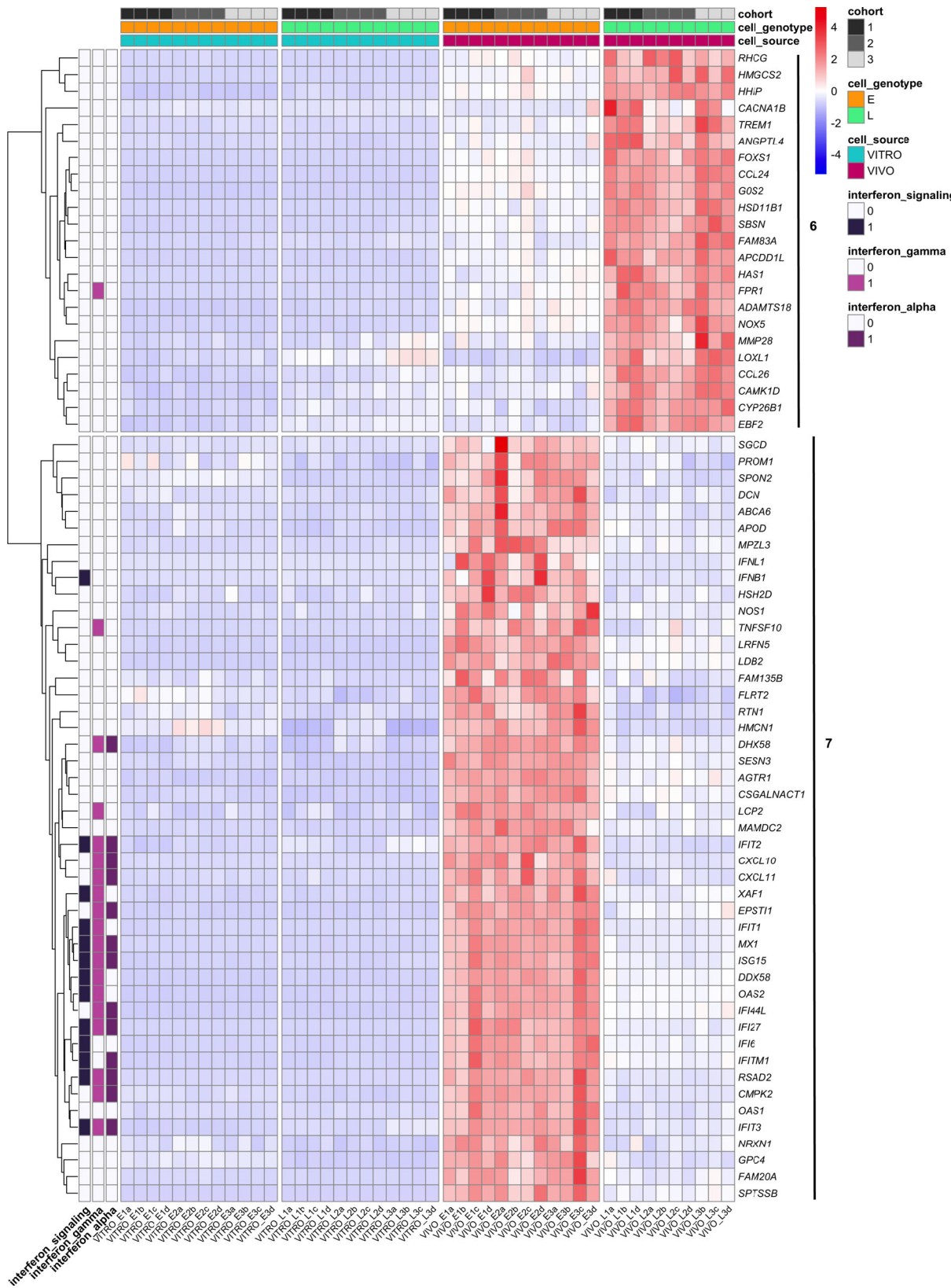

differentially expressed genes in vitro. A heatmap of the differentially expressed genes and GSEA using both Reactome and Hallmark datasets showed that the most significantly downregulated genes belonged to pathways involved in 'interferon (IFN) signalling' and interferon-alpha response'/'interferongamma response', with decreased expression of genes in these pathways correlating with worse survival in melanoma patients (Fig. 4 and Supplementary Figs. 7 and 8). Interestingly, IFNs have been demonstrated to counteract melanoma development through both cell-autonomous and non-cell-autonomous mechanisms, and IFN signalling is even more critical for suppressing metastatic melanoma spread[26]. A375_LRNmCherry cells also had strong

**Fig. 4 In vitro and in vivo differentially expressed genes between A375 melanoma cells expressing LRRN4CL or empty vector.** Cluster 6 and 7 of the heatmap of the differentially expressed genes (DEGs) between A375 melanoma cells over-expressing LRRN4CL (L) and A375 cells transduced with the empty vector (E) that were growing in vitro (VITRO) or in the lungs of mice for 21 days (VIVO). Supplementary Figure 7 shows the full heatmap with all eight clusters. Three independent experiments were performed (cohorts 1–3) with four samples (**a–d**) per experiment per cell line (two samples are not shown on the heatmap as they were outliers on the PCA). Genes were scored as differentially expressed if they had a $\log_2$ fold change $\leq -1$ or $\geq +1$ with a Padj < 0.01. Individual genes within gene set enrichment analysis (GSEA)-identified significantly down-regulated pathways are shown ('interferon_signaling', 'interferon_gamma', 'interferon_alpha').

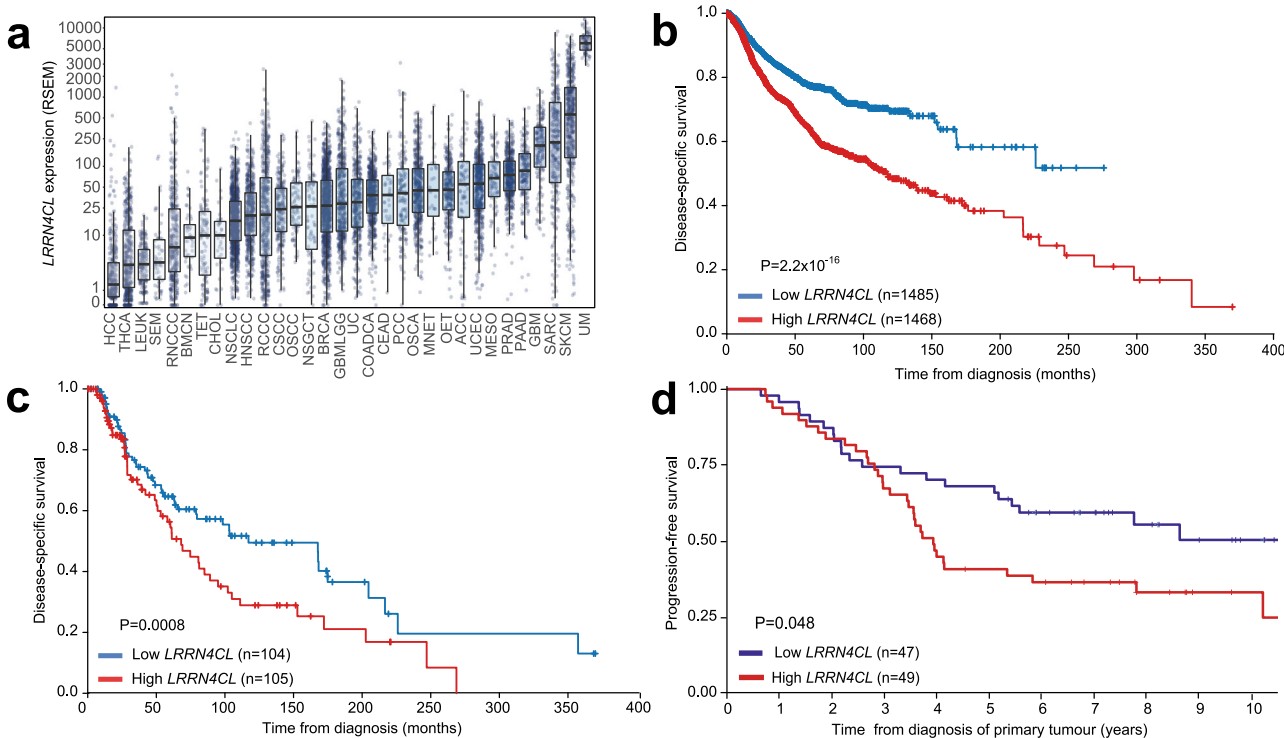

**Fig. 5 LRRN4CL expression in tumours and its correlation to patient outcome. a** *LRRN4CL* expression (RSEM) amongst different tumour types (data from TCGA). Bar represents the median, box represents the interquartile range. Whiskers represent 1.5 * IQR. Abbreviations: HCC hepatocellular carcinoma, THCA well-differentiated thyroid cancer, LEUK leukaemia, SEM seminoma, RNCCC renal non-clear cell carcinoma, BMCN mature B-cell neoplasms, TET thymic epithelial tumour, CHOL cholangiosarcoma, NSCLC non-small cell lung cancer, HNSCC head and neck squamous cell carcinoma, RCCC renal clear cell carcinoma, CSCC cervical squamous cell carcinoma, OSCC oesophageal squamous cell carcinoma, NCGCT non-seminomatous germ cell tumour, BRCA invasive breast carcinoma, GBMLGG diffuse glioma, UC bladder urothelial carcinoma, COADCA colorectal adenocarcinoma, CEAD cervical adenocarcinoma, PCC pheochromocytoma, OSCA oesophagogastric neuroepithelial tumour, MNET miscellaneous neuroepithelial tumour, OET ovarian epithelial tumour, ACC adrenocortical carcinoma, UCEC endometrial carcinoma, MESO pleural mesothelioma, PRAD prostate adenocarcinoma, PAAD pancreatic adenocarcinoma, GBM glioblastoma, SARC sarcoma, SKCM cutaneous melanoma, UM uveal melanoma. **b** Kaplan–Meier curves for disease-specific survival across all cancer patients from the TCGA dataset. Expression levels of *LRRN4CL* as lower 25th percentile ('low *LRRN4CL*') or upper 25th percentile ('high *LRRN4CL*'). Cox Log-rank *P* value, age- and sex-adjusted. **c** Kaplan–Meier curves for disease-specific survival in cutaneous melanoma patients from the TCGA dataset. Expression levels of *LRRN4CL* as lower 25th percentile ('low *LRRN4CL*') or upper 25th percentile ('high *LRRN4CL*'). Cox Log-rank *P* value, age- and sex-adjusted. **d** Kaplan–Meier curves for progression-free survival in cutaneous melanoma patients from the phase III adjuvant AVAST-M dataset[15]. Expression levels of *LRRN4CL* as lower 25th percentile ('low *LRRN4CL*') or upper 25th percentile ('high *LRRN4CL*'). Cox Log-rank *P* value, univariate analysis. The numbers in brackets on each graph indicate the number of patient samples per group.

in vivo upregulation of many genes with a known role in promoting metastasis. Examples include *LOXL1* (*lysyl oxidase-like 1*), with poorer survival of patients that have LOXL1-positive gastric cancer due to distant metastasis[27]; *MMP28* (*matrix metallopeptidase 28*), which is a promoter of invasion and metastasis in gastric cancer[28]; *HAS1* (*hyaluronan synthase 1*), which has been shown to play a role in the metastasis of breast cancer cells[29]; and *FPR1* (*formyl peptide receptor 1*), which is a cell-surface receptor expressed by highly malignant human glioma cells that mediates metastasis by regulating motility, growth, and angiogenesis through interactions with host-derived agonists[30]. Thus, upregulated LRRN4CL expression promoted lung-specific metastasis that was associated with microenvironment-stimulated cell-intrinsic

downregulation of IFN signalling and upregulation of pro-metastatic pathways. Interestingly, *LRRN4CL* is upregulated in glioma cells that express high levels of *MDA9*, which plays an important role in cancer progression, particularly during the invasion/metastasis stage[31,32]. The highest expression of *LRRN4CL* in the TCGA dataset was observed in cutaneous and uveal melanoma, both of which have a propensity to metastasise (Fig. 5a). Critically, high expression of *LRRN4CL* was significantly correlated with worse outcome in cancer patients, across all cancers represented in TCGA (Fig. 5b). High expression of *LRRN4CL* also correlated with poorer disease-specific survival in cutaneous melanoma patients (as assessed across two independent datasets; Fig. 5b, c), as well as for other cancer types (Supplementary Fig. 9).

Taken together with previous findings that *LRRN4CL* is not highly expressed in normal tissues and *Lrrn4cl* knockout mice show no overt phenotypes, we conclude that LRRN4CL represents an attractive therapeutic candidate for the prevention of metastatic colonisation.

## Methods

**Cell lines**. The mouse melanoma B16-F0 cell line was purchased from ATCC and the mouse melanoma B16-BL6 cell line was purchased from the University of Texas, MD Anderson Cancer Center; both were authenticated by whole genome and transcriptome sequencing[33]. The human melanoma A375 and MeWo cell lines were purchased from ATCC and verified by STR profiling. The other cell lines were gifts from the laboratories that generated them: the mouse melanoma HCmel12 cell line was from T. Tuting (University Hospital Magdeburg, Germany)[34], the mouse melanoma YUMM1.7 cell line was from M. Bosenberg (Yale University School of Medicine, USA)[35], the mouse colorectal MC-38 cell line was a gift from L. Borsig (University of Zurich, Switzerland)[36], the mouse mammary cancer EO771.LMB cell line was a gift from R. L. Anderson (Peter MacCallum Cancer Centre, Australia)[37], and the mouse bladder cancer MB-49 cell line was from A. Hegele (Philipps University of Marburg, Germany)[38]. The cell lines were grown in either DMEM (B16-F0, B16-BL6, HCmel12, YUMM1.7, MC-38, E0771 and A375) or RPMI (MB-49 and MeWo), with 10% (v/v) foetal calf serum (FCS) and 2 mM glutamine, 100 U/mL penicillin/streptomycin at 37 °C, 5% $CO_2$ (with the addition of 1% minimum non-essential amino acids for YUMM1.7 cells and 20 mM HEPES for EO771 cells). The cell lines were screened for the presence of mycoplasma and mouse pathogens (at Charles River Laboratories, USA) before culturing and never cultured for more than five passages.

**Mice**. Wildtype mice were C57BL/6NTac and immunodeficient mice were NOD.Cg*Prkdc^scid^, Il2rg^tm1Wjl^*/SzJ. Both lines were originally obtained from JAX Laboratories and maintained as core colonies at the Sanger Research Support Facility. *Lrrn4cl* 'knockout' (*Lrrn4cl^em1(IMPC)Wtsi^*) mice were generated at the Welcome Sanger Institute using CRISPR technology to disrupt the coding exon of *Lrrn4cl* using published methods[39] and phenotyped using an Ultrafocus100 densitometer (Faxitron). Heterozygous *Crtac1* (*C57BL/6NTac-Crtac1^em1(IMPC)H/H^*) mice were obtained from the MRC Harwell Institute which distributes these mice on behalf of the European Mouse Mutant Archive (www.emmanet.org)[40] and re-derived into the Sanger Institute Research Support Facility. The diet, cage conditions and room conditions of the mice were as previously detailed[41]. The care and use of all mice in this study were in accordance with the Home Office guidelines of the UK and procedures were performed under a UK Home Office Project License (P6B8058B0), which was reviewed and approved by the Sanger Institute's Animal Welfare and Ethical Review Body.

**Experimental metastasis assay**. The use of the experimental assay as a screen for pulmonary metastatic colonisation ability has been detailed previously[17]. For tail vein injections, the cells were resuspended in 0.1 mL phosphate-buffered saline (PBS) and injected into the lateral tail vein of 6- to 12-week-old wildtype or immunodeficient mice (both sexes were used but the mice were always of the same sex within an experimental cohort). The number of cells administered via the tail vein was dependent upon the cell line and experiment (detailed in the relevant figure legend). After 10 days (or 20–30 days for human cell lines) the mice were humanely sacrificed, their lungs removed and washed in PBS and the number of metastatic foci determined. As the B16-F0, B16-BL6 and HCmel12 cell lines produce pigmented metastases, their metastatic burden was determined by microscopic counting of the number of tumour foci on the surface of all five lobes of the lung by a single individual that was blind to the cell line administered to the mice. For all other cell lines, the lungs were fixed in 10% neutral-buffered formalin, paraffin-embedded, sectioned and haematoxylin and eosin (H&E) stained (as per routine histology techniques). The metastatic burden was determined by counting the number of tumour foci in one coronal section of all five lobes of lung by a pathologist that was blind to the cell line administered to the mice. For intrasplenic injections[41], the experiments were performed using wildtype mice and B16-F0 cells ($1 \times 10^4$). The metastatic burden was determined by counting the number of tumour foci in five sections of the liver. For all experimental metastasis assays, the raw data (number of metastatic foci counted in each mouse dosed with either the experimental or relevant control cell line) from each cohort of dosed mice was subjected to the non-parametric Mann–Whitney *t* test to determine significance.

**CRISPR activation membrane protein screen**. The mouse membrane protein ('m6') mCRISPRa-v2 subpooled library (consisting of 10,975 gRNAs targeting 2104 genes that encode for membrane proteins, and 250 non-targeting control gRNAs) was generated by Jonathan Weissman[10] and acquired from Addgene (#84003). The m6 plasmid library was packaged into lentivirus in HEK293T cells and transduced into a previously established B16-F0 mouse melanoma cell line stably expressing dCas9 ('dCas9-F0 cells'; the blasticidin-resistant dCas9 plasmid was a gift from Gavin Wright[42]). A total of $12 \times 10^6$ dCas9-F0 cells were transduced with the m6 library lentivirus at an MOI of 0.3 with 8 µg/mL polybrene. After 48 h, cells were

passaged ($16 \times 10^6$ cells re-seeded) and 5 µg/mL of puromycin added to the medium (flow cytometry was used to measure BFP expression and confirm successful transduction). After a further 4 days, cells were passaged again, maintaining $12 \times 10^6$ cells. On day 9, cells were detached, counted, centrifuged at 300g for 5 min then diluted in PBS. Aliquots of $5.5 \times 10^6$ and $1.1 \times 10^6$ cells (500× and 100× library representation, respectively) were pelleted and snap-frozen (representing '0-h' timepoint) and aliquots of $5.5 \times 10^5$ cells (50× library representation) in 100 µL PBS were intravenously administered (via tail vein) into 70 wildtype female mice aged 6–8 weeks. At two time points (4 h and 19 days post dosing), the mice were humanely sacrificed and saline cardiac perfused, with the lungs then being snap-frozen (labelled 'Lung_11' - 'Lung_45' for the 4-h cohort and 'Lung_46'–'Lung_80' for 19-day cohort). The lungs from each mouse (all five lobes) were homogenised in 1 mL Tris-buffered saline with 0.5% Triton X-100 and a portion taken for gDNA extraction (including 0-h timepoint dCas9-F0 cell pellets) using the Purgene kit (Qiagen) according to manufacturer's instructions. PCR reactions were performed with 500 ng of gDNA per reaction, using the Phusion® High-Fidelity PCR Master Mix with HF Buffer (NEB) to amplify the gRNAs. The forward primer contained an 8mer barcode, 5′ Illumina adapters and homology to the CRISPRia-v2 plasmid[10]. The reverse primer contained 3′ Illumina adapters and homology to the CRISPRia-v2 plasmid[10]. For each lung, 16 PCR reactions were performed and all products were pooled. A portion of this was purified to select for the desired ~280 bp product, using a Select-a-Size DNA Clean & Concentrator kit according to the manufacturer's protocol (Zymo). PCR was also performed on duplicate 0-h dCas9-F0 cell gDNA and the library plasmid DNA. Purity of all PCR samples was confirmed by analysis on a Bioanalyser. The samples were then combined in two pools, each containing 41 samples (1 plasmid sample, 2 cell line samples and 38 lung samples) and sequenced over 2 runs on a HiSeq2500 (Illumina). To ensure consistency between the 2 runs, 6 of the 70 lung samples in total were sequenced 'twice' with each duplicate being in a different pool/run to its original sample (specifically 'Lung_49b', 'Lung_56b', 'Lung_58b', 'Lung_66b', 'Lung_73b' and 'Lung_75b'). Two sequencing primers were used: a bespoke primer and a standard Illumina primer.

A screen was also performed using the EO771 mouse breast cancer cell line stably expressing dCas9 ('dCas9-EO cells'). The screen was performed as described above, except that the 70 mice were divided into three independent cohorts ('A' = 23 mice, 'B' = 23 mice, 'C' = 24 mice). Each cohort was tail vein dosed with independent replicates of $5.5 \times 10^5$ library-carrying dCas9-EO cells, and the mice were collected as and when they showed signs of pulmonary tumour burden, which was 18–28 days post dosing. In addition, three aliquots of $5.5 \times 10^6$ cells (500× library representation) were pelleted and snap-frozen per library replicate (representing '0-h' timepoint). The purified PCR products of all 79 samples (70 lung samples, labelled 'EOp_A1-A23', 'EOp_B1-B23' and 'EOp_C1-C24'; 3 sets of cell line samples in triplicate, labelled 'EOp_cells_A1, A2, A3', 'EOp_cells_B1, B2, B3', 'EOp_cells_C1, C2, C3') were pooled and sequenced over 1 run of a HiSeq2500 (Illumina). The analysis was performed using the 'percentile' method as described below, with all three cohorts analysed together.

**Bioinformatic analysis of the gRNAs**. Single-end reads (50 bp) were trimmed to remove adapter sequences and compared for exact matches against 11,225 sgRNAs from the Weissman murine membrane proteins (m6) library ('Top5' gRNAs including 250 non-targeting controls) [https://www.addgene.org/pooled-library/weissman-mouse-crispra-v2-subpools/]. Guides with fewer than 30 reads in either of the 'cell_500×' replicates were removed from the resulting count matrix. Two methods were used to identify enriched gRNAs: the 'percentile' method and the 'JACKS' method. For the 'percentile' method, total normalisation was performed with MAGeCK (version 0.5.8)[43], using 'cells_500×' samples as the control. gRNAs with no reads assigned were removed post-normalisation. For each gRNA at 19 d and 4 h, the count means, standard deviation, a number of samples in which the gRNA was present (group count) were generated. The standard error of the mean was calculated for each gRNA at 4 h and 19 d as the standard deviation divided by the square root of the group count. A per-gRNA *z*-score at 4 h and 19 d was determined as the normalised count minus the mean of the control counts, divided by the group SEM. gRNAs were ranked by *z*-score for each sample individually and gRNAs which were present in the 98th percentile in multiple samples identified. For the 'JACKS' method, a Bayesian analysis of the screen was performed using JACKS[11] (version: March 2018). Each sample was treated as an independent replicate, with all samples at the 4 h time point and all samples at the 19 d timepoint grouped together respectively, while the 'cells_500×' samples were used as the control. JACKS provides a gene-by-gene list of effect sizes and the standard deviations of the effect sizes for both the 19 d and 4 h samples. The effect of the gRNA on the metastatic potential of the cells was represented by both the difference of estimated JACKS effect sizes at 19 d compared to 4 h, and a 'z-score' which represents the difference of effect sizes divided by the standard deviation of the effect at 19 d.

**Validation of CRISPR activation screen 'hits'**. Individual gRNAs were cloned between the BstXI and BlpI sites of the CRISPRia-v2 plasmid using standard Weissman Lab protocols as recommended (https://weissmanlab.ucsf.edu/CRISPR/CRISPR.html). The gRNAs from the m6 library used for validation were 'Lrrn4cl_+_8850757.23-P1P2', 'Slc4a3_+_75546398.23-P1P2', 'Tm4sf19_-_32400563.23-P1P2' and

'non-targeting_01320–01324'. The plasmids were packaged into lentiviruses and dCas9-F0 cells were transduced and used in the experimental metastasis assay as detailed above.

**Generating *Lrrn4cl/LRRN4CL*-overexpressing cell lines**. For upregulated expression of LRRN4CL, the mouse or human *Lrrn4cl/LRRN4CL* cDNA (synthesised by GeneArt) was cloned into the multiple cloning site of the 'PB-CMV-MCS-EF1α-Puro PiggyBac' vector (PB510B-1) or 'PB-EF1α-MCS-IRES-Neo PiggyBac' vector (PB533A-2; System Biosciences). The stably-expressing *Lrrn4cl/LRRN4CL* cell lines were generated by co-transfection of the cells with 2 μg of either the *Lrrn4cl/LRRN4CL*-containing plasmids or 2 μg of the 'empty' plasmid, and 0.5 μg of PBase-expressing plasmid using Fugene HD (Promega) according to the man-ufacturer's recommendations. After selection in puromycin or G418 (Invivogen) for 7–10 days, the resulting colonies for each transfection were pooled to make the LRRN4CL expressing cell line ('LRN') or the control cell line ('PB'), with the cells being maintained in puromycin or G418 throughout.

To generated a 'tagged' version of LRRN4CL, an in-frame FLAG sequence and streptavidin sequence (216 bp) was included at the C-terminus of the mouse *Lrrn4cl* cDNA (synthesised by GeneArt) prior to the stop codon and cloned into the multiple cloning site of the PB510B-1 vector and a stably-expressing B16-F0 cell line (termed 'LRRN4CL-FSA' cells) generated as detailed above. For upregulated expression of LRRN4CL in a mCherry-expressing plasmid, human *LRRN4CL* cDNA was cloned into the multiple cloning site of the pReceiver-Lv206 plasmid (EX-NEG-Lv206; Genecopoeia). The plasmid was packaged into a lentivirus and used to transduce the A375 cell line (or the 'empty vector' virus was used to generate a 'control' cell line). After 48 h, the cells were placed in 2 μg/mL puromycin (which was maintained throughout their culturing). After 14 days in culture, cell sorting was performed (MoFlo XDP, Beckman Coulter) to select for those cells expressing the highest level of mCherry within the population.

**Cell growth assays**. Totally, $5 \times 10^4$ A375 cells were seeded per well of a 12-well plate. At 24, 48 and 72 h cells were harvested by trypsinisation and counted using trypan blue exclusion. The average of four wells was taken per technical replicate and three independent biological replicates of each cell line performed.

**Migration and invasion assays**. For migration assays, 24-well transwell chambers with 8.0 μm PET membranes were used and for invasion assays, Matrigel-coated invasion chambers with the same specification were used (Corning). A375 cells were serum-starved overnight before plating $2.5 \times 10^4$ cells per chamber in 200 μL serum-free DMEM medium. Wells were filled with 700 μL DMEM containing 10% FCS to encourage migration. Cells were incubated for 18 h for migration and 24 h for invasion assays. Following incubation, migrated cells attached to the reverse side of the membrane were fixed with methanol for 10 min and stained with Giemsa stain solution (Sigma). Three images were captured per technical replicate (in triplicate) using a light microscope with a camera (Leica Application Suite), and cells were counted manually using ImageJ (with three independent biological replicates of each cell line carried out).

**Glycosylation analysis**. Deglycosylation of whole-cell lysates from B16-F0 cells expressing a FLAG/streptavidin-tagged *Lrrn4cl* cDNA (LRRN4CL-FSA) was carried out with PNGase F (New England Biolabs) according to the manufacturer's instructions.

**Western blots**. Cells were lysed in RIPA lysis buffer (Merck Millipore) with the addition of phosphatase inhibitor (Sigma). Protein was quantified using a BCA assay and 50 μg protein loaded per well. Proteins were separated by SDS-PAGE and transferred to PVDF membrane. Membranes were blocked in 5% non-fat milk. Primary antibodies were as follows: Epithelial–Mesenchymal Transition Antibody Sampler Kit (note: expression of Slug was not detected in A375 cells; 1:1000 dilution, Cell Signalling #9782), Matrix Remodelling Antibody Sampler Kit (1:1000 dilution, Cell Signalling #73959), StemLight™ Pluripotency Transcription Factor Antibody Kit (note: expression of Nanog and Oct-4A was not detected in A375 cells; 1:1000 dilution, Cell Signalling #9093), LRRN4CL antibody—middle region (1:1000 dilution, Aviva Systems Biology, #OAAB0898), β-Actin (8H10D10) mouse monoclonal Antibody (1:2000 dilution, Cell Signalling #3700), α-Tubulin Antibody (1:1000 dilution, Cell Signalling #2144), and mouse monoclonal anti-FLAG® M2-Peroxidase (1:5000 dilution, Sigma #A8592). Secondary antibodies were as follows: anti-mouse IgG HRP-linked antibody (1:10,000, Cell Signalling #7076) and anti-rabbit IgG HRP-linked antibody (1:10,000, Cell Signalling #7074). The blots were visualised using chemiluminescence detection by the ImageQuant LAS 4000 machine (GE Healthcare). For all western blots, β-actin or α-tubulin were used as loading controls and run on the same gel as the corresponding protein of interest (membrane was either cut into sections for the different antibodies or stripped and re-probed with the loading control antibody).

**Gene expression**. RNA was extracted from tissues using an RNeasy fibrous kit (Qiagen) according to the manufacturer's instructions. *Crtac1* gene expression was assessed using FAM-conjugated TaqMan assays (Mm00513940_m1). Template

RNA was added in duplex reactions in triplicate with endogenous control *B2m* VIC primer limited probe (Mm00437762_m1), using QuantiNova Probe RT-PCR kit (Qiagen) with an Applied Biosystems Quantstudio 7 Flex Real-Time PCR system (ThermoFisher Scientific). Relative gene expression between endogenous control and target genes were analysed using the ΔΔCT method[44] with QuantStudio Software.

**Retrogenix cell microarray**. Primary and confirmatory screens with 10 μg/mL of a fusion protein consisting of the ECD of human LRRN4CL (amino acids 23–194) C-terminally linked to a human Fc domain (LRRN4CL-ECD-Fc; produced by Absolute Antibody) were carried out as detailed previously[45], based on probing an array of live human HEK293 cells over-expressing 5528 human transmembrane and cell-surface tethered soluble proteins. The positive control was the interaction between CD20 (one of the cDNAs within the library) and 1 μg/mL of Rituximab biosimilar antibody. The negative control was the use of PBS in place of the LRRN4CL-ECD-Fc. Detection of an interaction was visualised using Alexa-Fluor647 anti-human IgG Fc. For the flow cytometric validation assay, expression vectors encoding ZsGreen1 only, or encoding both ZsGreen1 and human *CRTAC1α*, human *CRTAC1β*, mouse *Crtac1* or *CD20* (positive control), were transfected, in duplicate, into human HEK293 cells. Live transfectants were incubated with 10 μg/mL LRRN4CL-ECD-Fc or 1 μg/mL Rituximab biosimilar positive control. Cells were washed and incubated with AF647 anti-human IgG Fc detection antibody as used in the cell microarray study. Cells were again washed, and ana-lysed by flow cytometry using an Accuri (Becton Dickinson). A 7AAD live/dead dye was used to exclude dead cells in the analysis, and ZsGreen+ (transfected) cells were selected for analysis.

**Preparation of lung cell suspensions**. Mice were perfused with 20 mL PBS by cardiac puncture. The perfused lungs were removed and disrupted in C tubes using the programme 'm_lung_01' with a gentleMACS (Miltenyi Biotec) in Hanks Balanced Salt solution containing calcium and magnesium. Liberase DL (Col-lagenase with low dispase content, Roche) was added to a final concentration of 0.1 U/mL and incubated for 30 min at 37 °C. The tubes were then processed using programme 'm_lung_02' and DNase (0.1 mg/mL) was added for a further 30 min at 37 °C. The resulting cell suspension was centrifuged at 400$g$ for 5 min, resus-pended in 2 mL fluorescence-activated cell sorting (FACS) buffer (D-PBS without calcium and magnesium-containing 2 mM EDTA, 0.5% FCS and 0.09% sodium azide), passed through a 30 μm cell strainer and analysed on the flow cytometer. To determine the number of melanoma cells present in the lungs of mice, the mice were tail vein dosed with $9 \times 10^5$ F0_LRN or F0_PB cells (pre-labelled with 10 μM CFDA (Molecular Probes, Invitrogen)) at 1 or 3 h before perfusion. An example of the gating strategy to detect CDFA+ tumour cells in the lung is shown in Sup-plementary Fig. 10, where an undosed mouse lung and the parental cell lines were used as negative and positive gating controls, respectively.

**Transcriptome sequencing**. Immunodeficient mice tail vein dosed with $2.5 \times 10^5$ A375_LRNmCherry or A375_EVmCherry cells were humanely sacrificed after 21 days and lung cell suspensions prepared using a human tumour dissociation kit (Miltenyi Biotec), followed by a mouse-cell depletion kit (Miltenyi Biotec), according to the manufacturer's instructions. Using a cell sorter (FACSAria Illu, Becton-Dickinson), A375_mCherry cells were identified after displaying in a bivariate plot of SSC-log versus mCherry by gating on high forward scatter versus side scatter to exclude some debris and dead cells and positively sorted. RNA was extracted from the sorted cells, as well as A375_LRNmCherry or A375_EVm-Cherry cells grown in vitro, using the RNeasy Mini kit (Qiagen), according to the manufacturer's instructions, and used to generate cDNA with the Smart-seq2 protocol[46,47]. Multiplexed sequencing libraries were generated from amplified cDNA using Nextera XT (Illumina). Multiplexed libraries were pooled and sequenced across multiple lanes on the Illumina HiSeq 2500 (V4). Paired-end 75 bp reads were aligned with STAR version 2.5.0c[48]. STAR genome index files were generated using a GTF file corresponding to gene models from Ensembl version 77 and reference genome version GRCh38. Read counting was performed with htseq-count from the HTSeq package (0.7.2)[49]. PCA identified two samples as outliers which were subsequently removed: VIVO_L1c and VIVO_L3a. Differential gene expression analysis was conducted with DESeq2 (version 1.26.0)[50]. After Benjamini-Hochberg correction, only genes with an adjusted $p$ value of <0.01 and a log$_2$ fold change $\geq 1$ or $\leq -1$ were considered to be significantly differentially expressed. Fast GSEA (FGSEA) was performed on 115 genes which were identified as being unique to the VIVO environment (LRRN4CL over-expression vs. empty vector) using the R package fgsea (version 1.12.0)[51] with the MSigDB (version 7.0) Hallmark and Reactome gene sets[52].

***LRRN4CL* expression levels in normal tissues, human tumours and survival**. Tissue RPKM expression data were extracted from the NCBI BioProject database (human data from PRJEB4337[12] and mouse data from PRJNA66167[53]). RPKM values for selected genes per tissue were extracted. Error bars show RPKM ± 1 standard deviation where available. *LRRN4CL* RSEM[54]-normalised RNA expres-sion data for all TCGA PanCancer Atlas Studies were downloaded from the cBioPortal[55,56]. These expression data are derived from the TCGA RNASeqV2

data. Expression data were filtered to remove any cancer types with fewer than matching 15 samples. The expression data were plotted using a logarithmic scale with a smooth transition to a linear scale close to zero to show low-expression data (using pseudo_log from ggplot2[57]). Sample survival data (both overall and disease-specific) were extracted from the TCGA PanCancer Atlas clinical data. LRRN4CL RSEM-normalised RNA expression data were extracted. Samples with LRRN4CL expression values in the top and bottom 25% were selected for analysis. Kaplan-Meier survival curves were created in R using the survival package (version 3.1-12)[58] for pan-cancer, cutaneous melanoma and uveal melanoma patient data and using OncoLnc[59] for a low-grade glioma, kidney renal clear cell carcinoma and ovarian carcinoma patient data. Cox Log-rank (Mantel–Cox) test p values were calculated. For the phase III adjuvant AVAST-M study[15], the clinical data acquired and the extracted RNA from the primary melanomas embedded in this study that were sequenced on the Illumina exome-capture sequencing platform were as previously described[60]. Kaplan–Meier survival curves were generated for the top and bottom 25% expression values as detailed above. The standardised LRRN4CL expression scores were then used as a continuous predictor in Cox regression models fitted by means of the coxph function of the survival package (v2.2-3) [https://CRAN.R-project.org/package=survival] in R (v3.5.1). The hazard ratio (95% CI) and p values corresponding to the signature were reported in both univariate and multivariate analyses. The clinical covariates stage, sex, age, number of involved regional lymph nodes, Eastern Cooperative Oncology Group performance status and treatment were considered in the multivariate survival model. Overall survival curves using the 13 differentially expressed genes from the RNAseq experiment that were in the Reactome 'interferon signalling' pathway was performed using GEPIA 2 [http://gepia2.cancer-pku.cn/#survival] on the 'SKCM' cohort from TCGA and statistics were calculated using the log-rank (Mantel–Cox) test.

**Statistics and reproducibility**. Statistical significance was assessed using either the non-parametric Mann–Whitney t test (for experimental metastasis assays) or the unpaired two-tailed t test (for in vitro assays). Statistical significance of differential gene expression in the transcriptomics data was assessed using DESeq2, correcting for multiple hypothesis testing using the Benjamini–Hochberg adjustment procedure. Statistical testing of the two groups displayed on the Kaplan–Meier survival curves was performed using the Log-rank test within the Cox's proportional hazards model, whilst correcting for clinical covariates as indicated.

**Reporting summary**. Further information on research design is available in the Nature Research Reporting Summary linked to this article.

## Data availability
The CRISPR data are available under the European Nucleotide Archive (ENA) accession number ERP123242 and the RNAseq data are available under the European Genome-phenome Archive (EGA) accession number EGAD00001006249. All source data for the figures are available in Supplementary Data 2. All relevant data are available from D.J.A. upon request.

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

## Acknowledgements

This work was supported by grants from Cancer Research UK (C20510/A13031 to D.J.A.), the Wellcome Trust (WT098051, to D.J.A.) and ERC Combat Cancer (319661 to D.J.A.). We would like to thank Mahrokh Nohadani (C&C Laboratories) and Yvette Hooks and Kirsty Roberts (Wellcome Sanger Institute) for the histological processing of the samples. We thank members of the CASM Support Lab (Wellcome Sanger Institute) for QC of the purified CRISPRa PCR products. We also thank members of the Research Support Facility (Wellcome Sanger Institute) for husbandry and ensuring the welfare of the mice and Mouse Pipelines (Wellcome Sanger Institute) for phenotyping of the *Lrrn4cl^{em1(IMPC)Wtsi}* mice. Finally, we thank Dr Tobias Bald (QIMR Berghofer Medical Research Institute) for his helpful discussions on the paper.

## Author contributions

Lvd.W. and A.O.S. devised the experiments. VH and GT performed all the viral work for the study. Lvd.W., GT and VH generated all the stably transfected cell lines for the study. Lvd.W. performed all of the tail vein dosing, subcutaneous dosing and experimental metastasis assays. Lvd.W. and A.S. measured tumour growth in the mice. A.C. and O.S. performed the intrasplenic dosing experimental metastasis assay. V.O. and V.I. performed the bioinformatic analysis for the CRISPRa screen. V.O. performed the bioinformatics analysis for the RNAseq dataset. A.D. performed the bioinformatics analysis for the TCGA datasets. R.R. performed the bioinformatics analysis on the AVAST-M dataset. V.H. performed all the in vitro metastatic capability assays. A.S. and V.H. performed the Western blots. M.J.A., M.T. and I.F. performed the histological analysis (counting of metastatic foci in all cell lines except B16 and HCmel12). M.P.C. and J.C. performed the deglycosylation experiments. A.O.S. performed all flow cytometry experiments and analysis. A.O.S. and A.S. extracted the RNA and performed the qPCR. Lvd.W., A.O.S. and D.J.A. led the project and wrote the paper, with contributions from all authors.

## Competing interests

The authors declare no competing interests.
