## [Peer Review File · Communications Biology]

Reviewers' comments:

Reviewer #1 (Remarks to the Author):

Weyden et al use an in vivo CRISPRa screen in a mouse melanoma model to identify LRRN4CL as a modulator metastasis. The authors show LRRN4CL does not modulate xenograft tumor cell proliferation and doesn't grossly regulate extravasation but instead seems to regulate IFN signaling and thereby growth/survival in lung mets. There are only a few small scale in vivo CRISPRa screens that I am aware of (PMC5930141, PMC4941480) and so this represents an emerging and really interesting area in functional genomics. Overall, I think this is an interesting and well-done study in part because to my knowledge using loss of function approaches it has been difficult to identify regulators of metastasis that do not also modulate primary tumor growth.

Minor points:

Any chance TM4SF19 or Slc4a3 have been shown to physically interact with LRRN4CL in a PPI database?

If you stimulate control or LRRN4CL overexpression cells in vitro with IFN is there a phenotypic difference?

It would be interesting to know whether knocking out IFN receptor or key signaling components would be epistatic to LRRN4CL overexpression in modulating metastasis.

Please remove "As there are no in vivo dCas9/CRISPRa screens in the literature" from the discussion due to (PMC5930141, PMC4941480).

Reviewer #2 (Remarks to the Author):

The manuscript by Weyden et al. describes the use of a genome-wide custom CRISPR (CRISPRa) activation library screening to identify membranous molecules responsible for metastasis to the lung. They identify *Lrrn4cl* as a key molecule.

Although this manuscript is very interesting, I have some concerns about the data presented.

Concerns

1. CRISPRa is useful technique for targeted gene overexpression; however, coactivators such as MS2-p65-HSF1 can make it more efficient. For example, how can a folded change be observed with gRNA for *Lrrn4cl* with this CRISPRa system? In addition, the efficiency of CRISPR activation generally depends on the targeted gRNAs. In theory, *Cxcr4* can be a candidate for this screenings. Can this library screen all targeted membranous molecules? This point should be considered.
2. Regarding Figure 3a, a positive control that can induce extravasation in 1 or 3 hours is missing. In addition, metastasis can generally be confirmed by microscopy analysis. Why have the authors not shown these types of data for metastatic sites?
3. The authors should show the background colony number for metastasis with dCas9-expressing B16-F0 in CRISPR screens, even if that number is very low.
4. The effect of overexpression of *Lrrn4cl* was confirmed in lung metastasis; however, knockout or knockdown of *Lrrn4cl* in the metastasis assay would also support the function of *Lrrn4cl* in high-*Lrrn4cl* expression cells.
5. The mechanism of *Lrrn4cl* in metastasis to the lung should be clarified. The authors focused on interferon alpha or gamma signaling in an in vivo context. If section assessment is possible, the interferon signals can be addressed by immunohistochemical analysis.
6. Why did the authors not use anti-LRRN4CL antibodies to prevent lung metastasis?
7. I agree that high expression of *LRRN4CL* might affect the prognosis of some cancers based on human database analysis; however, database analysis does not directly suggest lung

metastasis. The dataset should be reconsidered.

Minor points

1. "§" is missing for the first author.
2. The intent of the figures overlap for Fig.1b and Fig.1c. One of them can be supplementary.
3. Line 346 and 351: an extra reference index is observed.

Reviewer #1 (Remarks to the Author):

Weyden et al use an *in vivo* CRISPRa screen in a mouse melanoma model to identify LRRN4CL as a modulator metastasis. The authors show LRRN4CL does not modulate xenograft tumor cell proliferation and doesn't grossly regulate extravasation but instead seems to regulate IFN signaling and thereby growth/survival in lung mets. There are only a few small scale *in vivo* CRISPRa screens that I am aware of (PMC5930141, PMC4941480) and so this represents an emerging and really interesting area in functional genomics. Overall, I think this is an interesting and well-done study in part because to my knowledge using loss of function approaches it has been difficult to identify regulators of metastasis that do not also modulate primary tumor growth.

We thank the reviewer for their appreciation of the novelty of our study, with only a few small scale *in vivo* CRISPRa screen currently published, and that this is "*an emerging and really interesting area in functional genomics*". We also appreciate that they feel the study was "*well done*" and that they understand how difficult it is to identify regulators of metastasis that do not also affect growth of the primary tumour itself.

Minor points:

Any chance TM4SF19 or Slc4a3 have been shown to physically interact with LRRN4CL in a PPI database?

We have looked at several protein-protein interaction databases, and LRRN4CL has not been found to interact with either TM4SF19 or SLC4A3. LRRN4CL has only been reported to interact with the proteins shown below:

- BioGRID (v4.0; <https://thebiogrid.org/>) finds 5 physical interactors: FMA115C, GHITM, PIGA, PODXL2, and SRPK2, all determined through high-throughput screens using affinity capture mass spectrometry.
- STRING (v11.0; <https://string-db.org/>) finds 7 interactors: RWDD2A and SRPK2 (experimentally determined) and CACYBP, GANAB, EML3, HNRNPUL2 and INTS5 (determined by text-mining)

If you stimulate control or LRRN4CL overexpression cells *in vitro* with IFN is there a phenotypic difference?

We thank the reviewer for this suggestion. Since IFN β is considered the prototypical type I interferon, and *IFNB1* expression was strongly down-regulated *in vivo* in our LRRN4CL over-expressing cells (A375_LRN), relative to the control cells (A375_PB), we chose this IFN β to stimulate the cells *in vitro*, to identify a possible differential response between the A375_PN and A375_LRN cells. IFN β treatment of melanoma cells *in vitro* has been shown to result in growth inhibition and apoptosis [Garbe and Krasagakis, 1993; PMID: 7679433; Chawla-Sarkar et al., 2001; PMID: 11410525; Makita et al., 2019; PMID: 30864696]. Based on the *in vivo* data, we postulated that A375_LRN cells would be unresponsive to the elevated IFN β levels (as LRRN4CL over-expression induced strong down-regulation of cell intrinsic *IFNB1* expression) and thus would not show the same levels of growth inhibition or apoptosis as the A375_PB cells. Unfortunately, we were not able to show a difference between A375_PB and A375_LRN cells in terms of IFN β -induced growth inhibition (measured via the MTS assay after 72 hours exposure), early apoptosis (Annexin⁺/DAPI⁻ staining measured by FACS after 72 hours) or late apoptosis (measured by Annexin⁺/DAPI⁺ staining measured by FACS after 72 hours exposure), as shown below. These data suggest that LRRN4CL over-expression is only able have to a phenotypic effect

in vivo, specifically in the lung, possibly due to dependence upon some binding protein/co-factor and/or pulmonary microenvironmental conditions.

It would be interesting to know whether knocking out IFN receptor or key signaling components would be epistatic to LRRN4CL overexpression in modulating metastasis.

We considered such an experiment before submission of the manuscript, however, the biology of IFN receptors and their signalling is complex. The two IFN genes that were down-regulated in A375_LRN cells (relative to control cells) were *IFNL1* and *IFNB1*. *IFNL1* encodes the Interferon Lambda 1 (IFNL1) protein that functions via binding to the heterodimeric class II cytokine receptor composed of interleukin 10 receptor, beta (IL10R β) and Interferon Lambda Receptor 1 (IFNLR1, also known as IL-28 Receptor Subunit Alpha (IL28R α)). *IFNB1* encodes Interferon Beta 1 (IFN β) and signals mostly via binding to the Interferon Alpha And Beta Receptor Subunit 1-2 (IFNAR1-IFNAR2) heterodimeric receptor, but can also function with IFNAR1 alone. Since both *IFNL1* and *IFNB1* genes were down-regulated in A375_LRN cells, it is difficult to know which “IFN receptor” genes would be appropriate to knock out: IL10R β and IL28R α (to remove IFNL1 signalling) and/or IFNAR1 and IFNAR2 (to remove IFN β signalling). In addition, in the case of knocking out the IFNAR1-2 heterodimeric receptor, this would also affect IFN α signalling, which functions via binding to the same receptor, yet IFN α was not found to be down-regulated in our A375_LRN cells *in vivo*.

Similarly knocking out “key signalling components” is also complicated by the fact that whilst IFNL1 engagement with its receptor leads to the activation of the JAK/STAT signalling pathway resulting in the expression of IFN-stimulated genes (ISG), IFN β engagement with IFNAR1 alone (i.e., not complexed with IFNAR2) can result in signalling independent of JAK/STAT pathways. Furthermore, knocking out components of the JAK/STAT pathways would impact on the signalling abilities of a large number of receptors and thus it may not be possible to see the effect of LRRN4CL over-expression on this background. In addition, *Infar1*^{-/-} melanocytes (a cell line generated from a spontaneous skin lesion in a *Braf*^{V600E}; *Ifnar1*^{-/-} mouse) show aggressive primary tumour growth and metastasis to the lungs (when subcutaneously administered to

syngeneic mice) [Katlinskaya et al., 2016; PMID: 27052162], thus it may not be possible to delineate an additive effect of LRRN4CL over-expression on this background. In conclusion, the situation is complex and we believe it is not feasible to explore epistasis in this context.

We also have the complication that our Animal Facility is in the process of closing down (https://www.sanger.ac.uk/news_item/sanger-institute-animal-research-facility-close/) and thus breeding of stock lines has been reduced. This has been further exacerbated by COVID-19 induced lockdowns (only allowing work from home) and access restrictions (only allowing one animal technician per room). Thus unfortunately, the ability to do any large *in vivo* experiments has been severely curtailed. As such we respectfully request that such *in vivo* experiments be considered beyond the scope of the manuscript.

Although not requested by the reviewer, we would like to highlight the fact that we have included additional *in vivo* data in the revised manuscript. Specifically, whilst the manuscript was in review, we analysed a second *in vivo* CRISPRa screen, this time tail vein dosing EO771 mouse breast cancer cells, and our strongest ‘hit’ was *Lrrn4cl*. As a result, we have now included this screen in the current manuscript (mentioned at lines 150-157 in the Results section), as it supports our data from the B16-F0 screen, in identifying a role for *Lrrn4cl* in enhancing pulmonary metastatic colonisation.

Please remove “As there are no *in vivo* dCas9/CRISPRa screens in the literature” from the discussion due to (PMc5930141, PMc4941480).

We thank the reviewer for drawing our attention to these two papers, and have changed our sentence to read: “The CRISPRa system has been successfully used *in vivo* for screening pools of gRNAs targeting transcription start sites of 1-2 genes [Braun et al., 2016; Wangenstein et al., 2019], however, as yet there have been no large-scale *in vivo* screens performed using this system”. We hope this is acceptable.

Reviewer #2 (Remarks to the Author):

The manuscript by Weyden et al. describes the use of a genome-wide custom CRISPR (CRISPRa) activation library screening to identify membranous molecules responsible for metastasis to the lung. They identify *Lrrn4cl* as a key molecule.

Although this manuscript is very interesting, I have some concerns about the data presented.

We thank the reviewer for finding our manuscript “*very interesting*”.

Concerns

1. CRISPRa is useful technique for targeted gene overexpression; however, coactivators such as MS2-p65-HSF1 can make it more efficient. For example, how can a folded change be observed with gRNA for *Lrrn4cl* with this CRISPRa system? In addition, the efficiency of CRISPR activation generally depends on the targeted gRNAs. In theory, *cxcr4* can be a candidate for this screenings. Can this library screen all targeted membranous molecules? This point should be considered.

We have interpreted the reviewers question about the assay/experiment in several ways:

- (i) *We did not assess the specific fold change in LRRN4CL protein expression that was induced by the gRNA and co-activators:* We did attempt to do this with our anti-mouse LRRN4CL antibody (using Biomatik’s custom antibody production service; <https://www.biomatik.com/>) but did not get satisfactory results by Western blot; the antibody was able to detect the protein against which it was raised, but did

not produce any bands for the lanes using B16-F0 non-template control/*Lrrn4cl* gRNA cell lysates or FO_PB/LRN cell lysates (see below). However, we were able to use an anti-human LRRN4CL antibody (Supplementary Figure 3) to show that use of the human cDNA to over-express LRRN4CL showed a strong upregulation of the protein.

Western blot using **Biomatik anti-Lrrn4cl antibody**

- (ii) *Co-activators other than the ones we use are stronger:* It is true that co-activators such as MS2, p65 and HSF1 can make the activation of the transcription start site more efficient. However, it is unclear if supra-high levels of expression are physiological and our primary aim was to identify potential candidates, which could then be validated using an orthogonal assay. Specifically, the cDNA of *Lrrn4cl* was placed behind a ubiquitous promoter and as such over-expressed in the tumour cells, and we showed that these cells had the same phenotype as observed when the cells were expressing dCas9 and transduced with a gRNA targeting the TSS of *Lrrn4cl* (as shown in Figures 1b and 1c).

The reviewer is correct in stating that the efficiency of CRISPR activation is reliant on the gRNA itself. That is why the library we used contained 5 guides per gene, so as to target different TSSs of the gene. To this end, we cannot make a statement about the metastatic colonisation abilities of genes that were not a 'hit' in our screen (i.e., as per a 'drop out' CRISPR screen). We can only make statements about the metastatic colonisation abilities of genes for which the gRNA showed a phenotype. It is of course possible that other screening systems using different activators may yield additional hits and this is an area for future investigation.

As the reviewer notes, *Cxcr4* could indeed be a candidate for this type of screen. However, we did not screen for *Cxcr4* as it was not in our library. There are seven mouse sub-pooled CRISPRa-v2 sub-libraries commercially available (m1-m7) [Horlbeck et al., 2016; PubMed 27661255] and as stated in our materials and methods, we used the 'membrane proteins (m6)' sub-library, and so were limited to the 2,104 genes whose TSSs were targeted in this library. It is not comprehensive of all membrane proteins as some of the other sub-libraries (such as 'Kinases, Phosphatases and Drug Targets' (m1) and 'Cancer and Apoptosis' (m2), also have membrane proteins. Indeed, gRNAs against the transcription start sites of *Cxcr4* are found in the 'cancer and apoptosis' (m2) sub-library.

2. Regarding Figure 3a, a positive control that can induce extravasation in 1 or 3 hours is missing. In addition, metastasis can generally be confirmed by microscopy analysis. Why have the authors not shown these types of data for metastatic sites?

The Figure 3a data show the comparison of the levels of extravasation of LRRN4CL over-expressing cells (“LRN”) compared to those seen with the PB cells, and there was no overt difference between the two, thus we stated “This suggests that the expression of *Lrrn4cl* cDNA did not enhance the ability of the tumour cells to extravasate”. We do not have a “positive control” tumour cell line that shows enhanced levels of extravasation in 1-3 hours. Our control in this experiment is the mice dosed with ‘PB’ cells (i.e., cells transfected with the empty vector backbone only). We are not sure of the reviewers concerns with this assay/experiment, however, we have considered several possible scenarios:

- (i) *The cells may not have completed the extravasation process at 1-3 hours post-dosing.* To this end, we know that some cells have completed the extravasation process by virtue of the fact that we can detect the presence of the fluorescently-labelled tumour cells in the lung at these timepoints, which demonstrates that the assay is working correctly. Similarly, other studies have tail vein dosed mice with fluorescently labelled cells and analysed the lungs by flow cytometry only 30 mins post-dosing (Tichet et al., 2015, Nature Communications, PMID: 25925867). Indeed, studies performed by the world-renowned metastasis expert, the late Professor Isiah Fidler, showed that after tail vein injection of 1×10^5 radio-labelled B16-F1 cells, $64,000 \pm 6,000$ cells can be detected in the lungs of the mice at 2 minutes post-administration (compared to 170 ± 80 cells in the blood (from 0.5mL sampled/mouse). After that timepoint they continued to decrease in number, with the lungs showing $57,000 \pm 4,200$ cells at 1 hour, $32,700 \pm 2,700$ cells at 3 hours and $1,190 \pm 200$ cells at 24 hours [Fidler and Nicholson, 1976; PMID: 1003551].
- (ii) *We may not be able to detect any differences in extravasation between the 2 cell lines at these timepoints.* Xiong et al., 2020 [PMID: 32015106] showed that Twist-expressing and Hsp47-expressing MCF10A breast cancer cells showed statistically significantly enhanced number of cells in the lungs of mice only 4 hours post-dosing (relative to control MCF10A cells), whereas silencing of Hsp47 in these cells resulted in a significant reduction of those cells in the lung at 4 hours. In addition, we have previously used this assay to show that *Spns2*^{-/-} mice have no difference from wildtype mice in the number of B16-F10 cells in their lungs 30 minutes after tail vein dosing, yet by 10 days they have significantly fewer pulmonary metastatic colonies than the wildtype mice (which we later proved was due to elimination of the metastatic cells via NK cells and T cells in the lungs of the *Spns2*^{-/-} mice) [van der Weyden et al., 2017, Nature, PMID: 28052056]. Thus, we are confident that use of this assay and these timepoints is able to detect differences in extravasation abilities, should they exist, and that no difference between the cells in these experiments is correctly informing us that *Lrrn4cl*-over-expressing cells do not have an advantage in terms of enhanced extravasation abilities.

The reviewer is entirely correct that metastasis can be confirmed by microscopical analysis. Indeed, we sectioned and H&E-stained the lungs of mice dosed with a range of cell lines and the number of metastatic colonies/foci was ascertained by a pathologist using light microscopy (MC-38, EO771.LMB and MB-49 cells in Figure 2b; A375 and MeWo cells in Figure 2c). This was also done using B16-F0 cells when counting hepatic metastatic colonisation (Figure 2f). However, it is not a viable option to use this technique at timepoints as early as 1-3 hours post-dosing, as it would require serial sectioning of the entire lung (generating over 1000 sections to be analysed for extravasated single tumour cells) in order to be able to ‘see’ a sufficient number of cells (hence why analysing by FACs is more sensitive). In addition, when our pathologist looked at the H&E sections of lungs of mice at earlier timepoints post-dosing (compared to the 10 day timepoint when we usually count the number of metastases), he

reported that “At 5 days, the melanoma tumours are so small (mostly less than clusters of 5 cells which I used as the threshold for reliable & reproducible tumour recognition) that it is unreliable to count the tumour numbers. At 10 days, there is a clear difference in metastatic melanoma tumour number. The conclusion here is that 5 days is too early to collect lung metastatic tumours as they are not yet established as identifiable tumour clusters/nodules”. Thus, to try counting at 1-3 hours post-dosing is not feasible. To use microscopy to visualise cells at this early stage of metastasis, would ideally involve intravital imaging, however, we do not have access to such specialised equipment, nor are we allowed to perform such experiments on our Home Office Licence (it would require a Project amendment that would be unlikely to be granted given no one at our Institute has the expertise in performing such experiments).

3. The authors should show the background colony number for metastasis with dCas9-expressing B16-F0 in CRISPR screens, even if that number is very low.

We have previously presented these data and shown in our own laboratory/animal facility that B16-F0 cells have a very low metastatic colonisation ability relative to their more metastatic derivatives, B16-F10 and B16-BL6 cells [Figure 1b; Del Castillo Velasco-Herrera et al., 2018; PMID: 29193607], which is in agreement with the seminal experiments performed by Professor Isiah Fidler (as described above in point (2) above).

4. The effect of overexpression of *Lrrn4cl* was confirmed in lung metastasis; however, knockout or knockdown of *Lrrn4cl* in the metastasis assay would also support the function of *Lrrn4cl* in high-*Lrrn4cl* expression cells.

We anticipated that a reviewer may ask us to generate *LRRN4CL* knockout cells to confirm the phenotypic findings of *LRRN4CL* over-expression cells, and we started to generate CRISPR knockout *LRRN4CL* cell lines at the time of submission of the manuscript. Details are as per below:

Generation and use of *LRRN4CL*-deficient human melanoma cell lines. *LRRN4CL* gRNA sequences (GCACTTCTCCCATGCGCGG, ‘guide 1’ and CGTAGTCGCAGGGGACAGC, ‘guide 2’) or safe-targeting control gRNA sequences (GATCAGGGAATCTTTGAGAA and GTGAAGATGATCGCTTATAC) [Morgens et al., 2017] were cloned into gRNA expression vector pKLV2-U6gRNA5(BbsI)-PGKpuro2ABFP-W (Addgene #67974) using T4 DNA Ligase (NEB) and the resulting three vectors used to produce lentivirus. Blasticidin-resistant A375 and MeWo cells stably expressing Cas9 [Thompson et al., 2020] were infected with the addition of 8 µg/mL polybrene. After 48 hours cells expressing gRNAs were selected using puromycin (2 µg/mL). For the *LRRN4CL* gRNA cell lines, 18 individual colonies (per guide) were picked 8 or 14 days later, for A375 and MeWo cells, respectively, and expanded. For the safe-targeting control gRNA cell lines, all the colonies on the plate were pooled at 8 or 14 days after selection started, for A375 and MeWo cells, respectively, and expanded. To characterise the CRISPR-induced indels DNA was extracted from the cell lines using the Purgene kit (Qiagen) according to manufacturer’s instructions, and the region surrounding the gRNA site was amplified by 3 rounds of nested PCR: round 1 used the primers, ExtF: CCA CTA CGC AAA CGA CAT AAA and ExtR: TTC TGA TCC CCA GTT GCA TT to generate a 500 bp product, round 2 used the primers, MiSEQ_IntF: ACA CTC TTT CCC TAC ACG ACG CTC TTC CGA TCT CTG GAG AGT CCT GGG CAC and MiSEQ_IntR: TCG GCA TTC CTG CTG AAC CGC TCT TCC GAT CTT TCC CTT CCA GTC TCC ATG C to generate a 300 bp product, and round 3 used primers to add on the barcode and Illumina adapters. The pooled PCR products were sequenced on a MiSeq (Illumina) using 150 bp paired-end reads. Each A375 cell line sample had an average of 35,000 reads and each MeWo cell line sample had an average of 4,000 reads.

Identification of the indels was performed by aligning the sequencing result for each clone with the sequence for *LRRN4CL*. Cell lines (n=2-3) that showed disruption of both *LRRN4CL* alleles were expanded for use in the experimental metastasis assay (as well as the pooled control cell lines). NOD-SCID mice (6-8 weeks of age) were tail vein dosed with 2.5×10^5 A375 cell clones or 0.75×10^5 MeWo cell clones. After 20 days, the lungs were fixed in 10% neutral-buffered formalin before being paraffin-embedded, sectioned and haematoxylin and eosin (H&E) stained (as per routine histology techniques), and the metastatic burden determined by counting the number of tumour foci in one coronal section of all 5 lobes of lung (performed by a pathologist who was blind to the cell line administered to the mice).

Despite all the clones we chose carrying indels that resulted in a shift of the reading frame of both alleles, the tail vein assay results were not consistent. As shown below (with symbols representing individual mice), some clones showed significantly decreased pulmonary metastases (A375_2.1, A275_2.17 and MeWo_1.17), yet others showing unchanged metastases (A375_1.3, MeWo_1.23) or increased metastases (A375_1.11). As we cannot explain the different phenotypes between these clones, we cannot say for certain that loss of *LRRN4CL* results in decreased pulmonary metastatic colonisation.

Although not requested by the reviewer, we would like to highlight the fact that we have included additional *in vivo* data in the revised manuscript. Specifically, whilst the manuscript was in review, we analysed a second *in vivo* CRISPRa screen, this time tail vein dosing EO771 mouse breast cancer cells, and our strongest 'hit' was *Lrrn4cl*. As a result, we have now included this screen in the current manuscript (mentioned at lines 150-157 in the Results section), as it supports our data from the B16-F0 screen, in identifying a role for *Lrrn4cl* in enhancing pulmonary metastatic colonisation.

5. The mechanism of *Lrrn4cl* in metastasis to the lung should be clarified. The authors focused on interferon alpha or gamma signaling in an *in vivo* context. If section assessment is possible, the interferon signals can be addressed by immunohistochemical analysis.

Our RNAseq experiment showed that interferon beta and lambda expression was decreased in *LRRN4CL* over-expressing cells growing in the lung (*in vivo*), with *INFB1* and *IFNL1* both showing ~5-fold down-regulation relative to control cells (Figure 4 and Supplementary Table 3). We did not report on altered expression in interferon alpha or gamma signalling in these cells. Since interferon beta (IFN β) is the canonical type I IFN family member, we chose to use IFN β for assessing interferon signals by immunohistochemical analysis. The antibody was from Thermo

(Cat #PA5-20390), with detection performed using Vector lab Vectastain Elite ABC kit (PK-6101). However, as shown below, it was not possible to determine if the A375-LRN cells showed increased IFN β levels over A375-PB cells due to the presence of heavy background/non-specific staining.

NSGM138.2d – Cohort 1, A375-PB

NSGM154.1a – Cohort 1, A375-LRN

NSGM211.1c – Cohort 2, A375-PB

NSGM201.1b – Cohort 2, A375-LRN

It is quite likely this may be due to the fact that we had to use sections from formalin-fixed, paraffin-embedded (FFPE) blocks from experiments when we used H&E-stained sections of the lung to count the number of metastasis. As such, the lungs were sitting in the fixative for many days before being processed, and thus not the ideal situation when using the tissue for IHC. However, due to our Animal Facility closing down next year (https://www.sanger.ac.uk/news_item/sanger-institute-animal-research-facility-close/), many of the stock lines have been wound down/removed from the shelf, and COVID-19 pandemic means access to the Animal Facility is limited (to ensure compliance with social distancing regulations and protect the animal staff so they are able to remain at work to care for the animals). Thus we are not able to perform new experiments to collect 'fresh' lungs from dosed mice for IHC staining. We hope the reviewer can be sympathetic with the current situation that is making additional *in vivo* experiments near impossible.

6. Why did the authors not use anti-LRRN4CL antibodies to prevent lung metastasis?

We considered such an experiment before submission of the manuscript however, all anti-human LRRN4CL antibodies commercially available are only *in vitro* grade and not suitable for *in vivo* application, so it is not possible to administer them to the mice. In addition, they are not necessarily 'blocking/neutralising' antibodies, being generated solely for Western blotting applications. We did go to the effort of generating an anti-mouse LRRN4CL antibody however, that was also *in vitro* grade, and only suitable for Western blotting.

7. I agree that high expression of *LRRN4CL* might affect the prognosis of some cancers based on human database analysis; however, database analysis does not directly suggest lung metastasis. The dataset should be reconsidered.

The reviewer is absolutely correct that the database analysis does not directly suggest that high expression of *LRRN4CL* results in worse prognosis due to lung metastasis. However, there are no appropriate datasets available to test this suggestion – we performed a wide search for transcriptomic datasets of human melanoma for patients for which it was known whether or not they developed lung metastases, and could not find any (there is nothing available in the literature or on ArrayExpress or cBioPortal (TCGA)). We also contacted Professor Julia Newton-Bishop, who is a Professor of Dermatology and an expert in melanoma research and factors moderating melanoma survival (with over 180 publications) and she said she was not aware of any such dataset.

All we can say is that for melanoma patients it is accepted that metastasis (and rarely the primary tumour itself) is the cause of death for most patients, with the lung being a common site of metastasis. We predict that a patient carrying a primary melanoma with increased *LRRN4CL* expression would have a high chance of developing lung metastases, as opposed to liver metastases. Adding weight to this is the fact that increased *LRRN4CL* expression in uveal melanoma patients did not correlate with poor survival (Supplementary Fig. 9b) and uveal melanoma almost exclusively metastasises to the liver, not the lung (and we showed that elevated *LRRN4CL* levels did not confer any selective growth/survival advantage in the liver; Fig. 2d). We have now mentioned this in the manuscript (lines 252-255).

Minor points

1. "\$" is missing for the first author.

This was intentional. It was to recognise that the second and third authors gave equal contributions to the manuscript, but unfortunately only one could be in second position.

2. The intent of the figures overlap for Fig.1b and Fig.1c. One of them can be supplementary.

It is true that the intent of the figures are the same, in that they demonstrate over-expression of *LRRN4CL* resulting in increased metastatic pulmonary colonisation abilities of the melanoma cells. However, there are subtle differences, in that Figure 1b uses B16-F0 cells that have been lentivirally co-transduced with a plasmid carrying dCas9 and a plasmid carrying a gRNA against the transcription start sites of one of several different genes (including *Lrrn4cl*). In contrast, Figure 1c uses B16-F0 cells that have been co-transfected with a plasmid carrying Piggybac transposase (PBase) and a plasmid carrying the *Lrrn4cl* cDNA. Thus, we would respectfully like to keep both of them as part of Figure 1.

3. Line 346 and 351: an extra reference index is observed.

Thank you for picking these up. They have now been removed.

REVIEWERS' COMMENTS:

Reviewer #1 (Remarks to the Author):

The authors have satisfied all of my concerns. Inclusion of the EO771 screen data is a nice addition.

Reviewer #2 (Remarks to the Author):

The authors, Weyden et al., have addressed my concerns very well. The knockout experiments are very interesting, and the LRRN4CL-KO results suggest a decrease in the number of lung metastases. Regarding the dataset for melanoma, I totally agree that a large study has not yet been performed; however, some datasets have information about TNM classifications (M1b; lung metastasis), such as GSE19234. Alternatively, lymphatic metastasis could be discussed. I also totally agree again that performing immunohistochemistry of secretory proteins in FFPE (formalin-fixed paraffin-embedded) samples is very difficult. Therefore, I was thinking that estimation of the downstream molecules, such as the phosphorylation status of STAT1, could be done using frozen samples. However, if not available, then expression profiles are acceptable. Potentially, all experiments would be unstable to some extent, therefore, I was thinking that, ideally, negative/positive controls could be performed in each experiment. However, I believe that this CRISPR activation screening has revealed a new mechanism for lung metastasis of malignant melanoma and the manuscript is suitable for the Communications biology.

Response to reviewers (COMMSBIO-20-1992A)

Reviewer #1:

The authors have satisfied all of my concerns. Inclusion of the EO771 screen data is a nice addition.

We are delighted that the reviewer is satisfied with the changes we made at their request and they appreciate the addition of the results from the EO771 cell line screen.

Reviewer #2:

The authors, Weyden et al., have addressed my concerns very well. The knockout experiments are very interesting, and the LRRN4CL-KO results suggest a decrease in the number of lung metastases.

We are pleased that the reviewer is happy that all their concerns have been addressed and that they appreciate our efforts to generate and phenotype *LRRN4CL* KO cell lines.

Regarding the dataset for melanoma, I totally agree that a large study has not yet been performed; however, some datasets have information about TNM classifications (M1b; lung metastasis), such as GSE19234.

The reviewer is correct that there are datasets that contain information related to site of metastases, however, these datasets are few and far between and are also very limited in the number of suitable cases they contain. For example, in the case of the GSE19234 dataset mentioned, there are only 3 patients with a status of M1b (indicating lung metastases), which is obviously insufficient to be able to draw any conclusions. In addition, this study was performed on the Affymetrix Human Genome U133 Plus 2.0 Array platform, and this array does not have probes covering *LRRN4CL* (ENSG00000177363). Another example is the study by Winnepenninckx and co-workers [PMID: 16595783], which has gene expression profiles and follow up data for 83 patients with cutaneous melanoma, however, only 9 patients had metastases and none of them were to the lung (5 were to other skin sites and 4 were to regional lymph nodes). Thus, the type of data that we need to be able to draw any firm conclusions is not available.

Alternatively, lymphatic metastasis could be discussed.

We are not sure what the reviewer wishes us to discuss about lymphatic metastasis. Our experimental design was set up to look at (i) haematogenous dissemination, not lymphatic dissemination, of tumour cells (as we used tail vein administration of the tumour cells), and (ii) pulmonary metastases, not lymph node metastases (as we only collected the lungs from the mice, not lymph nodes).

I also totally agree again that performing immunohistochemistry of secretory proteins in FFPE (formalin-fixed paraffin-embedded) samples is very difficult. Therefore, I was thinking that estimation of the downstream molecules, such as the phosphorylation status of STAT1, could be done using frozen samples. However, if not available, then expression profiles are acceptable.

We thank the reviewer for appreciating our attempt to perform IHC for IFN β on FFPE sections. Unfortunately we do not have any frozen lung samples from our experiments as the tissues were routinely put into formalin. Thus we hope our expression profiles (Figure 4) are acceptable.

Potentially, all experiments would be unstable to some extent, therefore, I was thinking that, ideally, negative/positive controls could be performed in each experiment.

For the type of experiment that was performed in Figure 3a, we typically only include *negative* controls (specifically, we used the lung from a mouse that had *not* been dosed with any fluorescently labelled tumour cells) as it is hard to justify dosing a cohort of mice with a *positive* control (i.e., cells that we know will show increased or decreased levels of metastasis); it is not within the recommended guidelines on use of laboratory animals in the UK to perform a regulated procedure on an animal with the intention that the results are for use as a positive control.

However, I believe that this CRISPR activation screening has revealed a new mechanism for lung metastasis of malignant melanoma and the manuscript is suitable for the Communications biology.

We believe that both reviewers consider our manuscript to be suitable for publication in Communications Biology, and have made the necessary revisions to ensure we comply with the formatting requirements.